# When Semantic Segmentation Meets Frequency Aliasing

**Linwei Chen[1], Lin Gu[2,3] & Ying Fu[1]** *
[1]Beijing Institute of Technology, Beijing, China
[2]RIKEN AIP, Tokyo, Japan
[3]The University of Tokyo, Tokyo, Japan
`chenlinwei@bit.edu.cn    lin.gu@riken.jp    fuying@bit.edu.cn`

## Abstract

Despite recent advancements in semantic segmentation, where and what pixels are hard to segment remains largely unexplored. Existing research only separates an image into easy and hard regions and empirically observes the latter are associated with object boundaries. In this paper, we conduct a comprehensive analysis of hard pixel errors, categorizing them into three types: false responses, merging mistakes, and displacements. Our findings reveal a quantitative association between hard pixels and aliasing, which is distortion caused by the overlapping of frequency components in the Fourier domain during downsampling. To identify the frequencies responsible for aliasing, we propose using the equivalent sampling rate to calculate the Nyquist frequency, which marks the threshold for aliasing. Then, we introduce the aliasing score as a metric to quantify the extent of aliasing. While positively correlated with the proposed aliasing score, three types of hard pixels exhibit different patterns. Here, we propose two novel de-aliasing filter (DAF) and frequency mixing (FreqMix) modules to alleviate aliasing degradation by accurately removing or adjusting frequencies higher than the Nyquist frequency. The DAF precisely removes the frequencies responsible for aliasing before downsampling, while the FreqMix dynamically selects high-frequency components within the encoder block. Experimental results demonstrate consistent improvements in semantic segmentation and low-light instance segmentation tasks. The code is available at: `https://github.com/Linwei-Chen/Seg-Aliasing`.

## 1 Introduction

Semantic segmentation is a crucial task in computer vision, with numerous applications such as medical segmentation (Falk et al., 2019), autonomous driving (Hu et al., 2023), robotics (Milioto & Stachniss, 2019), and urban planning (Liu et al., 2023; Chen et al., 2021; 2022a; Fu et al., 2022a). This dense prediction task heavily relies on high-frequency spatial information that encompasses fine details like textures, patterns, structures, and boundaries (Zhang et al., 2019; Liu et al., 2021a; Zhu et al., 2021; Ding et al., 2019).

While recent models (Cheng et al., 2022; Wang et al., 2023) have achieved much progress, most of these existing methods treat all pixels equally and average each pixel's loss for the image-level one. However, it has long been observed that some pixels are harder to segment than others (Li et al., 2017; Gu et al., 2020). Previous works (Gu et al., 2020; Deng et al., 2022) observe that pixels prone to error are associated with object boundaries, poor lighting conditions, *etc.* Considering this, (Li et al., 2017; Wang et al., 2020) separates an image into two parts: easy and hard regions according to the confidence of predicted probability before applying different segmentation heads. Still, quantitative analysis of 'hard pixels' is an open and challenging problem (Gu et al., 2020).

In this paper, we analyze hard pixels at boundaries and categorize them into three types: 1) false responses, 2) merging mistakes, and 3) displacements. As illustrated in Figure 1, false responses occur when the model predicts boundaries in areas without the object of interest. Merging mistakes

---
*Corresponding Author

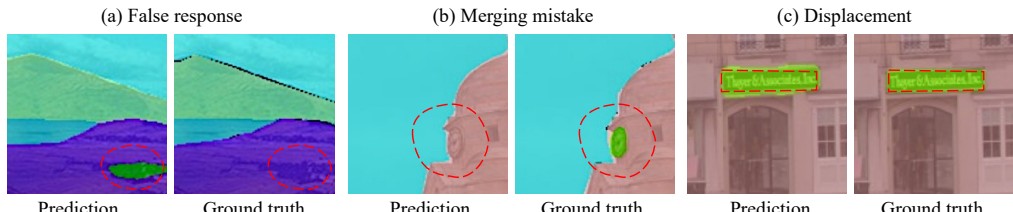

Figure 1: Illustration of three boundary error types.

refer to the failure to predict boundaries in regions containing the object of interest, resulting in the erroneous merging of two objects into one or missing predictions. Displacements involve predictions that are close to the correct location but deviate from the ground truth object boundary.

We further quantitatively associate three types of errors with a previously unexplored phenomenon: aliasing. Aliasing refers to the overlapping of frequency components in the Fourier domain when a signal is undersampled. According to the Nyquist-Shannon Sampling Theorem (Shannon, 1949; Nyquist, 1928), signals with frequencies *higher than half of the sampling rate* will be distorted. This is a particularly critical issue in semantic segmentation networks, which typically employ a series of downsampling operations to expand the receptive field and reduce dimensionality (He et al., 2016; Liu et al., 2021b; Wang et al., 2023; Cheng et al., 2022; Chen et al., 2022b).

Measuring aliasing level needs to calculate the Nyquist frequency. Existing research (Grabinski et al., 2022) calculates Nyquist frequency based on the downsampling stride, *e.g.*, a stride of 2 assumes a sampling rate of $\frac{1}{2}$, causing frequencies above $\frac{1}{4}$ (half of the sampling rate) to become aliased. This assumption holds for point-wise downsampling or downsampling without channel expansion, *e.g.*, a 1×1 convolution, max pooling. However, for larger kernels (*e.g.*, 2×2) and wider output channels (*e.g.*, 2× wider), the actual downsampling rate should be larger, *i.e.*, $\frac{\sqrt{2}}{2}$.

In light of this, we propose to calculate the actual Nyquist frequency based on an equivalent sampling rate, *i.e.*, equivalent Nyquist frequency. The determination of this equivalent Nyquist frequency depends on both the sampling kernel size and the size of the input-output feature maps. Then, we measure the aliasing level with an aliasing score, which is defined as the ratio of high-frequency power above our Nyquist frequency to the total power of the spectrum.

As shown in Figure 9, our analysis reveals a positive correlation between the hard pixels at boundaries and the aliasing score. All three types of errors increase as the aliasing score rises. These errors exhibit distinct characteristics when analyzed from the perspective of aliasing. Additionally, we also note that the importance of these errors varies across different scenarios. For example, displacement errors are primarily concentrated in areas with a high aliasing score, as depicted in Figure 9. In critical tasks such as robotic surgery or radiation therapy, even a displacement of only two or three pixels from vital organs like the brainstem or the main artery can have fatal consequences. In contrast, false responses and merging mistakes, which hold greater significance in autonomous driving scenarios, tend to be more prevalent in areas with relatively low aliasing scores.

The aliasing-induced error can partially explain the success of low-pass filter-based methods (Zhang, 2019; Zou et al., 2020; Grabinski et al., 2022; Chen et al., 2023), as they reduce specific high-frequency components to alleviate aliasing. But they either empirically set the low-pass kernel or underestimate the frequency threshold. Here, we propose two simple yet effective solutions: the de-aliasing filter (DAF) and the frequency mixing module (FreqMix). Based on our calculation of the equivalent Nyquist Frequency, DAF accurately removes the frequencies responsible for aliasing in the Fourier domain during specific downsampling (*e.g.*, 3×3 convolution with stride of two). Given the importance of high-frequency components in representing crucial detailed information (Li et al., 2020a; Bo et al., 2023), FreqMix can dynamically select and balance high-frequency components in each encoder block (*e.g.*, ResNet block). Our main contributions can be summarized as follows:

- Going beyond the binary distinction of easy and hard regions, we categorize pixels challenging for segmentation into three types: false responses, merging mistake, and displacements. These three error types hold varying levels of significance in different tasks.

- We introduce the concept of equivalent sampling rate for the Nyquist frequency calculation and propose an aliasing score for quantitative measurement of aliasing levels. Our metric effectively characterizes the three types of errors with distinct patterns.

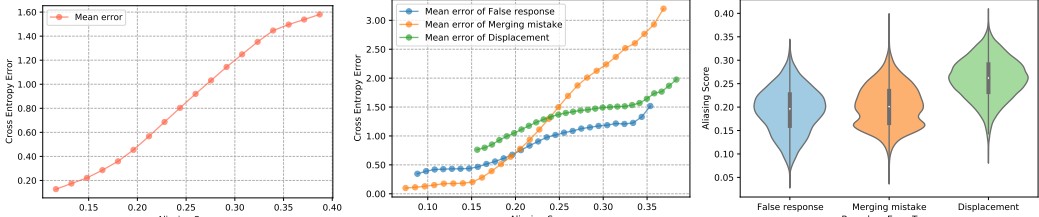

Figure 2: Left: The correlation curve between the cross-entropy error of boundary pixels and the aliasing score. Middle: Cross-entropy error curve for three types of boundary pixels, correlated with aliasing score. Right: Distribution of three types of hard pixels.

- We design a simple de-aliasing filter to precisely remove aliasing as measured by our aliasing score. Additionally, we propose a novel frequency-mixing module to dynamically select and utilize both low and high-frequency information. These modules can be easily integrated into off-the-shelf semantic segmentation architectures and effectively reduce three types of errors. Experiments demonstrate consistent improvements over state-of-the-art methods in standard semantic segmentation tasks and low-light instance segmentation.

## 2 RELATED WORKS

**Hard pixels.** Early researches (Shrivastava et al., 2016; Lin et al., 2017b) have found significant variations in classifying samples within images for object detection. Online Hard Example Mining (OHEM) (Shrivastava et al., 2016) selects challenging samples with high losses during training and ignores easily classifiable samples. In contrast, Focal Loss (Lin et al., 2017b) adaptive assigns weights to challenging samples according to their losses. These strategies show effectiveness. Some semantic segmentation approaches (Li et al., 2017) also employ hard sample mining. For instance, Li *et al*. (Li et al., 2017) use shallow or deep networks based on predicted probability scores for easy or hard regions, respectively. Another study (Wang et al., 2020) uses three segmentation heads for coarse segmentation. They process hard and easy regions based on predicted scores and merge the results for the final output. Additionally, methods like Online Hard Region Mining (OHRM)(Yin et al., 2019) and NightLab(Deng et al., 2022) identify challenging regions based on loss values (Yin et al., 2019) or a detection network (Deng et al., 2022). In this work, our finding reveals a positive correlation between the hard pixels and the aliasing score, which is previously unexplored.

**Aliasing in Neuron Networks.** The aliasing effect in neuron networks is receiving increasing attention. The very first attempt (Zhang, 2019) identified that it is the aliasing makes the neuron network sensitive to the shift. (Zhang, 2019) also proposed the anti-aliasing filters on the convolution layers to enhance shift-invariance. (Zou et al., 2020) introduces learned blurring filters to further improve shift-invariance. Recently, (Hossain et al., 2023) suggests combining a depth adaptive blurring filter with an anti-aliasing activation function. Wavelets transformation (Li et al., 2021) also offers a solution by leveraging low-frequency components to reduce aliasing and enhance robustness against common image corruptions. Beyond image classification, anti-aliasing techniques have gained relevance in the domain of image generation. In (Karras et al., 2021), blurring filters are utilized to remove aliases during image generation in generative adversarial networks (GANs), while (Durall et al., 2020) and (Jung & Keuper, 2021) employ additional loss terms in the frequency space to address aliasing issues. Unlike the empirical design of low-pass filters in existing works, we quantitatively calculate the frequency threshold for aliasing and precisely remove the corresponding high frequencies during downsampling.

**Frequency learning** Rahaman *et al*. (Rahaman et al., 2019) find the spectral bias that neuron networks prioritize learning the low-frequency modes. Xu *et al*. (Xu & Zhou, 2021) conclude the deep frequency principle, the effective target function for a deeper hidden layer biases towards lower frequency during the training (Xu & Zhou, 2021). Qin *et al*. (Qin et al., 2021) exploring utilizing more frequency for channel attention mechanism. Xu *et al*. (Xu et al., 2020) introduces learning-based frequency selection method into well-known neural networks, taking JPEG encoding coefficients as input. Huang *et al*. (Huang et al., 2023) employs the conventional convolution theorem in DNN, demonstrating that adaptive frequency filters can efficiently serve as global token mixers. Compared to existing works, this work investigates the relationship between high frequency and segmentation

Figure 3: Illustration of how a larger kernel and channel expansion increase the sampling rate. Left: Point-wise downsampling with a stride of 2 and no channel expansion results in a sampling rate of $\frac{1}{2}$, causing aliased high frequencies to be misrepresented as low frequencies. Right: Downsampling using $2 \times 2$ identity kernels with $4\times$ channel expansion actually increases the sampling rate to 1, instead of $\frac{1}{2}$, as all pixels are sampled.

error for the first time with introduced aliasing score, demonstrating the possibility of improving segmentation accuracy by optimizing the frequency distribution of features.

## 3 ALIASING DEGRADATION AND SOLUTION

### 3.1 ALIASING MEASUREMENT

Irrespective of specific architecture, modern Deep Neural Networks (DNNs) fundamentally consist of a series of stacked convolutional or transformer encoder blocks and downsampling layers (He et al., 2016; Liu et al., 2021b; 2022). From the perspective of signal processing, the downsampling operations are prone to violate Nyquist Sampling Theorem (Shannon, 1949; Nyquist, 1928) since the frequency of features, especially those close to the boundary, often exceeds the Nyquist frequency. The resulting frequency overlaps, formally known as aliasing, occur when high frequencies above the Nyquist frequency are undersampled, leading to signal distortion or misinterpretation.

**Nyquist frequency calculation.** To calculate the Nyquist frequency during downsampling, existing research (Grabinski et al., 2022) considers the stride of the downsampling layer. Specifically, for a downsampling layer with a stride of 2, since it reduces the spatial size by half, the sampling rate is $\frac{1}{2}$. According to the Nyquist Sampling Theorem (Shannon, 1949), the Nyquist frequency is half of the sampling rate, *i.e.*, $\frac{1}{4}$. This estimation is accurate for $1\times1$ point-wise downsampling layers or depth-wise downsampling layers without channel expansion, such as $1\times1$ convolutions, max/average pooling. However, it may underestimate the sampling rate when downsampling layers employ larger kernel sizes and output channel expansion, as they increase the actual sampling rate, such $2\times2$ patch merging and $3\times3$ convolution in (Liu et al., 2022; 2021b; Wang et al., 2023).

To illustrate this phenomenon, we provide a simple example in Figure 3. By learning four different identity kernels and outputting $4\times$ wider output channels, even though the spatial size is $2\times$ down-sampled, the spatial information can be losslessly preserved because all pixels are actually sampled, *i.e.*, the sampling rate is 1 instead of $\frac{1}{2}$.

Based on this observation, we consider the kernel size and feature size (channel, height, and width) instead of just the downsampling stride. We introduce an alternative yet straightforward solution, equivalent sampling rate (ESR), to calculate the actual sampling rate as:

$$\text{ESR} = \min(K^{\text{down}}, \sqrt{\frac{C^{\text{out}}}{C^{\text{in}}}}) \times \sqrt{\frac{H^{\text{out}} \times W^{\text{out}}}{H^{\text{in}} \times W^{\text{in}}}}, \tag{1}$$

where $C$, $H$, and $W$ represent channel, height, and width sizes. 'in' and 'out' refer to input and output features. We assume equal sampling rates for height and width, calculated by square root, *i.e.*, $\sqrt{\frac{H^{\text{out}} \times W^{\text{out}}}{H^{\text{in}} \times W^{\text{in}}}} = \frac{1}{\text{Stride}}$, which aligns with (Grabinski et al., 2022). $\min(K^{\text{down}}, \sqrt{\frac{C^{\text{out}}}{C^{\text{in}}}})$ reflects the impact of downsampling kernel size $K^{\text{down}}$ and channel expansion. For example, with a $2\times2$ kernel ($K^{\text{down}} = 2$), it can double the sampling rate if the channel expands by a factor of 4 ($\frac{C^{\text{out}}}{C^{\text{in}}} = 4$), accommodating the sampled information. However, with lower channel expansion (*e.g.*, $\frac{C^{\text{out}}}{C^{\text{in}}} = 2$), it limits the sampling increment to $\min(2, \sqrt{2}) = \sqrt{2}$. Specifically, ResNet (He et al., 2016), Swin-Transformer (Liu et al., 2021b), and ConvNeXt (Liu et al., 2022) use a $3\times3$ residual block (can be regarded as $3\times3$ convolution (Ding et al., 2021)), $2\times2$ patch merging with a $2\times$ channel expansion. Thus, the equivalent sampling rate is $\frac{\sqrt{2}}{2}$, and the corresponding Nyquist frequency is $\frac{\sqrt{2}}{4}$.

**Aliasing score.** After calculating the Nyquist frequency, we can obtain the set of high frequencies larger than the Nyquist frequency that lead to aliasing: $\mathcal{H} = \{(k, l) \mid |k| > \frac{\text{ESR}}{2} \text{ or } |l| > \frac{\text{ESR}}{2}\}$. Thus, we can quantitatively measure the aliasing level by introducing an aliasing score as a metric,

which we define as the ratio of frequency power above the Nyquist frequency to the total power in the frequency spectrum.

Specifically, we first transform the feature map $f \in \mathbb{R}^{C \times H \times W}$ into the frequency domain using the Discrete Fourier Transform (DFT), it can be represented as:

$$F(c, k, l) = \frac{1}{H \times W} \sum_{h=0}^{H-1} \sum_{w=0}^{W-1} f(c, h, w) e^{-2\pi j (kh + lw)}, \tag{2}$$

where $F \in \mathbb{R}^{C \times H \times W}$ represents the output array of complex numbers from the DFT. $H$ and $W$ denote its height and width. And the $c$, $h$, $w$ indicates the coordinates of feature map $f$. The frequencies in the height and width dimensions are given by $|k|$ and $|l|$, with $k$ taking values from the set $\{0, \frac{1}{H}, \ldots, \frac{H-1}{H}\}$ and $l$ from $\{0, \frac{1}{W}, \ldots, \frac{W-1}{W}\}$. Consequently, the aliasing score can be formulated as follows:

$$\text{Aliasing Score} = \frac{\sum_{(k,l) \in \mathcal{H}} |F(c, k, l)|^2}{\sum |F(c, k, l)|^2}. \tag{3}$$

## 3.2 BOUNDARY ERROR TYPE

Although it is widely known that predicting object boundaries accurately is challenging and often results in high errors, there is no existing work that investigates boundary error types in-depth. Here, we further divide the boundary error into three types, false response, missing edges, and displacement, as shown in Figure 1. Through aliasing analysis, we find they show different characteristics.

**False response** refers to instances where the predicted edges erroneously appear in non-boundary areas, also known as false positives. These false responses are often associated with texture and noise-related issues as reported in previous studies (Lopez-Molina et al., 2013). Notably, as shown in Figure 9, these errors tend to occur in regions characterized by a relatively low aliasing score.

**Merging mistake** occurs when the predicted results fail to capture the corresponding edges, resulting in false negatives. They are typically linked to low-contrast regions (Lopez-Molina et al., 2013). As depicted in Figure 9, these errors also manifest in areas with a relatively low aliasing score.

**Displacement** error arises when the predicted edges deviate from their true positions. Such displacements may occur when the image features are misaligned (Li et al., 2020b; Huang et al., 2021). As depicted in Figure 9, these errors tend to manifest in areas with a relatively high aliasing score.

**Metrics for three errors.** Additionally, we have designed a series of metrics as tools to assess and analyze the error rate of three distinct types of hard pixel errors at boundaries: false response (FErr), merging mistake (MErr), and displacement (DErr):

$$\text{FErr} = \frac{|P_d - (G_d \cap P_d)|}{|P_d|}, \text{MErr} = \frac{|G_d - (P_d \cap G_d)|}{|G_d|}, \text{DErr} = 1 - \frac{|(P_d \cap P) \cap (G_d \cap G)|}{|P_d \cap G_d|} \tag{4}$$

where $P$ and $G$ refer to the binary mask of prediction and ground truth, $P_d$ and $G_d$ represent the pixels located in the boundary region of the binary mask with a pixel width of $d$. As per the guidelines presented in (Cheng et al., 2021), we set the pixel width of the boundary region to 15 pixels.

**Impact of blur filters**. Previous work (Zhang, 2019) has explored the use of a simple Gaussian blur to mitigate aliasing in DNNs. However, due to the absence of a metric for evaluating the level of aliasing and the oversight of three distinct types of hard pixel errors at boundaries, the reason why the counterintuitive

Table 1: Analysis for the impact of blur filters with UPerNet-R50 on Cityscapes validation set. The ↑/↓ indicates that higher/lower values are better.

| Blur Kernel | mIoU↑ | Boundary | | Three type errors | | | Aliasing Score |
|---|---|---|---|---|---|---|---|
| | | BIoU↑ | BAcc↑ | FErr↓ | MErr↓ | DErr↓ | |
| - | 78.1 | 61.8 | 74.4 | 27.2 | 25.1 | 26.9 | 9.4% |
| 3×3 | **78.8** | **62.3** | **75.0** | **26.6** | **24.7** | **26.5** | 0.27% |
| 5×5 | 78.7 | 62.2 | 74.8 | 26.8 | 25.0 | 26.8 | 0.16% |
| 7×7 | 78.2 | 61.7 | 74.5 | 27.1 | 25.3 | 27.2 | 0.14% |

phenomenon of a simple blur filter improving performance remains unexplained. Here, armed with the tools of aliasing scores and metrics for three distinct types of hard pixel errors, we take a step further. As shown in Table 1, inserting a 3×3 Gaussian filter before downsampling effectively reduces the average aliasing score from 9.4% to 0.27%, thus improving both the overall segmentation mIoU and boundary segmentation BIoU.

Further increasing the blur kernel size to 5×5 and 7×7 does not yield additional improvements; instead, it leads to degradation compared to the 3×3 kernel. This occurs because the larger kernel sizes blur boundary details, resulting in higher error rates for false responses (FErr), merging

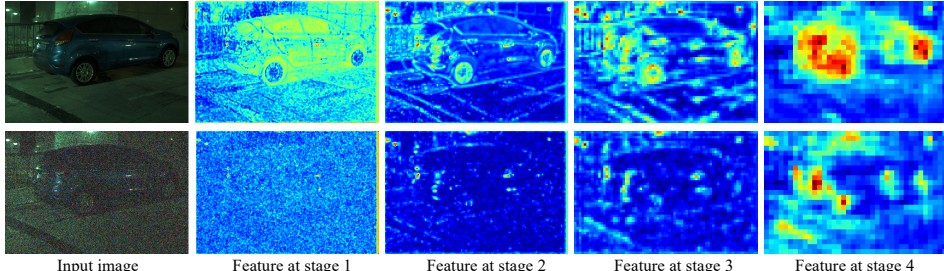

| Input image | Feature at stage 1 | Feature at stage 2 | Feature at stage 3 | Feature at stage 4 |

Figure 4: Illustration of the impact of noise on the features at different stages.

mistakes (MErr), and displacement errors (DErr). In summary, an appropriately sized blur kernel improves the model by reducing hard pixel errors. However, larger blur kernels can result in a higher error rate across three types of errors, potentially counteracting the improvements

**Impact of noise.** Previous work (Chen et al., 2023) has revealed the negative impact of noise in low-light scenarios. Here, we further analyze the relationship between aliasing and noise. Modern backbone models typically organize the convolutional/transformer encoder block into four stages. The first stage is downsampled by a factor of 4 compared to the input images, and there are three $2\times$ downsampling operations from stage-1 to stage-4. Consequently, we present the aliasing score results for the three $2\times$ downsampling operations at stages 1 to 3.

As shown in Table 2, we introduce Gaussian noise to images and observe a substantial increase in the aliasing ratio of the feature map at the first stage. This observation indicates that noise amplifies the severity of alias-

Table 2: Analysis for the impact of noise with UPerNet-R50 on Cityscapes validation set.

| Noise level | mIoU↑ | Boundary | | Three type errors | | | Aliasing score | | |
|---|---|---|---|---|---|---|---|---|---|
| | | BIoU↑ | BAcc↑ | FErr↓ | MErr↓ | DErr↓ | Stage-1 | Stage-2 | Stage-3 |
| $\sigma = 0$ | **78.1** | **57.4** | **73.0** | **25.4** | **53.3** | **27.6** | 5.4% | 11.5% | 11.3% |
| $\sigma = 5$ | 71.2 | 55.2 | 66.1 | 34.1 | 33.8 | 33.9 | 6.1% | 11.0% | 10.8% |
| $\sigma = 10$ | 54.1 | 39.7 | 50.4 | 47.3 | 49.0 | 45.7 | 6.4% | 10.3% | 9.9% |
| $\sigma = 20$ | 20.6 | 15.0 | 22.0 | 72.7 | 78.7 | 71.2 | 9.6% | 8.4% | 6.4% |

ing effects. However, for deeper features at the second and third stages, the aliasing ratio decreases. This decrease does not signify a reduction in aliasing; rather, it occurs because the earlier high level of aliasing results in severe degradation. Consequently, the features become hard pixels, meaning that the following stage struggles to extract useful information and loses response to objects. Thus feature maps appear plain and lack useful high frequency, as shown in Figure 4. As a result, there is a significant deterioration in all three types of errors, particularly in the case of displacement error.

### 3.3 ALIASING-AWARE SOLUTION

In this section, we introduce two solutions: De-Aliasing Filter (DAF) for removing aliasing during downsampling and Frequency Mixing (FreqMix) for adjusting frequency within encoder block.

**De-aliasing filter.** Several prior techniques, as introduced in (Zhang, 2019; Zou et al., 2020; Hossain et al., 2023), employ methods to attenuate high-frequency components within feature maps prior to downsampling, aiming to prevent aliasing artifacts. These approaches typically employ traditional spatial domain blurring operations for this purpose. Recent work (Grabinski et al., 2022) also explores the removal of high frequencies in the Fourier domain. However, it determines the low-pass cutoff frequency solely

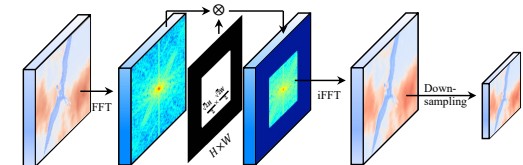

Figure 5: Illustration of de-aliasing filter (DAF). FFT: Fast Fourier Transform, iFFT: Inverse FFT.

based on the downsampling stride, leading to an underestimation of the sampling rate. This results in an excessive removal of high frequencies that are crucial for predicting segmentation details.

On the basis of our Nyquist frequency calculation in Section 3.1, we propose the De-Aliasing Filter (DAF). It first transforms the feature map to be downsampled into the Fourier domain and then sets the power of high frequencies that lead to aliasing to zero. Leveraging the equivalent sampling rate equation 7, we can calculate the actual Nyquist frequency, which serves as the low-pass cut-off frequency for the DAF. It accurately removes the frequency power causing aliasing. This precise removal of frequency power effectively eliminates aliasing. Figure 5 illustrates its procedure in detail. In the spatial domain, this operation would involve convolving the feature map with an infinitely large non-bandlimited filter, which cannot be implemented in practice (Grabinski et al., 2022). Note that DAF is a parameter-free and easily integrable module.

**Frequency Mixing Module.** The degradation introduced by aliasing underscores the importance of frequency balancing in segmentation. On one hand, high frequencies are crucial for representing important detailed information, such as object boundaries (Li et al., 2020a; Bo et al., 2023). On the other hand, high frequencies above the Nyquist frequency can lead to harmful aliasing. Motivated by this observation, we shift our focus to the feature extraction blocks and introduce a frequency-mixing module. We insert this module after the convolutional layer, where it adaptively weights the frequency power both below and above the Nyquist frequency. Its formal representation is:

$$f'(c, h, w) = A^\downarrow(c, h, w) * f^\downarrow(c, h, w) + A^\uparrow(c, h, w) * f^\uparrow(c, h, w), \quad (5)$$

where $c$, $h$, and $w$ are indices in the channel, height, and width dimensions, and $f$ and $f'$ are input and output feature maps. $f^\downarrow$ and $f^\uparrow$ are frequency components below and above the Nyquist frequency, which can be simply obtained by low-pass/high-pass filtering in the Fourier domain, as shown in Figure 6. $A^\downarrow$ and $A^\uparrow$ are the corresponding weighting values for frequency mixing.

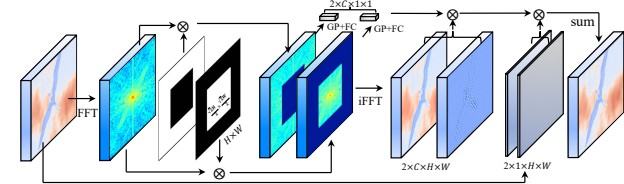

Figure 6: Illustration of Frequency Mixing (FreqMix) module.

However, directly predicting $A^\downarrow$, $A^\uparrow \in \mathbb{R}^{C \times H \times W}$ can be computationally heavy. Thus, we decompose the weighting values prediction into channel-wise and spatial-wise components

$$f'(c, h, w) = A_1^\downarrow(c) * A_2^\downarrow(h, w) * f^\downarrow(c, h, w) + A_1^\uparrow(c) * A_2^\uparrow(h, w) * f^\uparrow(c, h, w), \quad (6)$$

where the channel-wise prediction can be simply implemented by global pooling followed by a fully connected layer, and spatial-wise prediction can be implemented by a convolution layer. We use the sigmoid function to regularize the weighting values to be in the range $[0, 1]$. FreqMix can easily integrate into the encoder block of existing models (He et al., 2016; Wang et al., 2023).

## 4 EXPERIMENTS

### 4.1 METRIC AND DATASET

**Metric.** Following previous works (Long et al., 2015; Cheng et al., 2022), we employ standard metrics like mean intersection-over-union (mIoU). For a finer evaluation of boundary segmentation accuracy, inspired by (Cheng et al., 2021), we use mean boundary IoU and accuracy (BIoU, BAcc). Additionally, we introduce FErr, MErr, and DErr from equation 4 to assess distinct hard pixel errors at boundaries: false responses, merging errors, and displacements.

**Datasets.** For semantic segmentation, we employ widely-used and challenging Cityscapes (Cordts et al., 2016), PASCAL VOC (Everingham et al., 2010), and ADE20K (Zhou et al., 2017) datasets. Cityscapes (Cordts et al., 2016) consists of 5,000 finely annotated images, each with dimensions of $2048 \times 1024$ pixels. This dataset is meticulously divided into 2,975 images for the training set, 500 for the validation set, and 1,525 for the testing set. It

Table 3: Ablation study for low-pass cut-off frequency on the Cityscapes validation set. A higher low-pass cut-off frequency allows more high-frequency components to pass through.

| Cut-off frequency | mIoU↑ | Boundary | | Three type errors | | | Aliasing Score |
|---|---|---|---|---|---|---|---|
| | | BIoU↑ | BAcc↑ | FErr↓ | MErr↓ | DErr↓ | |
| - | 78.1 | 61.8 | 74.4 | 27.2 | 25.1 | 26.9 | 9.4 |
| $\frac{1}{4} \times 1.0$ | 78.6 | 62.7 | 74.9 | 26.7 | 24.8 | 26.7 | 0.0 |
| $\frac{1}{4} \times 1.1$ | 78.7 | 62.3 | 75.4 | 26.7 | 24.8 | 26.1 | 0.0 |
| $\frac{1}{4} \times 1.2$ | 78.8 | 62.4 | 75.4 | 26.3 | 24.3 | 26.2 | 0.0 |
| $\frac{1}{4} \times 1.3$ | 79.2 | 62.4 | 75.4 | 26.2 | 24.3 | 26.1 | 0.0 |
| $\frac{1}{4} \times \sqrt{2}$ | **79.3** | **62.6** | **75.7** | **26.0** | **24.0** | **25.9** | 0.0 |
| $\frac{1}{4} \times 1.5$ | 78.9 | 62.4 | 75.0 | 26.6 | 24.6 | 26.4 | 1.51 |
| $\frac{1}{4} \times 1.6$ | 79.1 | 62.1 | 75.2 | 26.7 | 24.5 | 26.5 | 3.17 |

encompasses 19 distinct semantic categories. Pascal VOC (Everingham et al., 2010) involves 20 foreground classes and a background class. After augmentation, it has $10,582/1,449/1,456$ images for training, validation and testing, respectively. ADE20K (Zhou et al., 2017) is a notably challenging dataset that encompasses a whopping 150 semantic classes. This extensive dataset features a distribution of 20,210 images for training, 2,000 for validation, and 3,352 for the test set. Considering the importance of low-light applications (Zhang et al., 2021; 2023; Fu et al., 2022c), which are often characterized by significant noise (Fu et al., 2022b; Wei et al., 2020; 2021; Zou & Fu, 2022; Zou et al., 2023), we also use the low-light instance segmentation dataset LIS (Hong et al., 2021; Chen et al., 2023) to evaluate the proposed method. It provides 8 common classes and consists of 2,230 pairs of low-light and normal-light images, with 70% for training and 30% for testing.

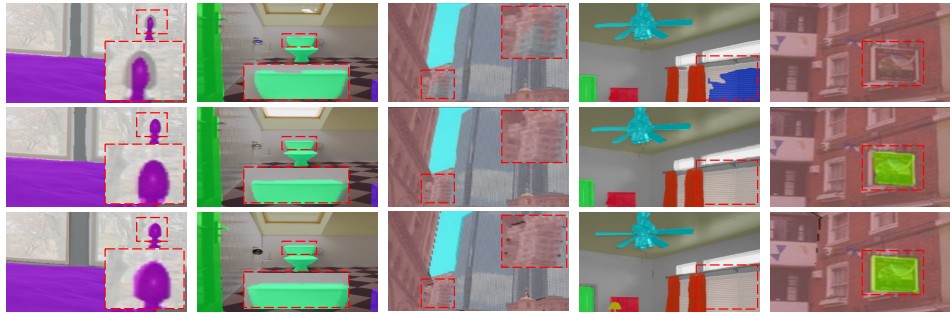

Figure 7: Visualized results on ADE20K. The top and middle rows represent predictions without and with the proposed methods, respectively. The bottom row represents the ground truth.

## 4.2 IMPLEMENT DETAILS

All UPerNet (Xiao et al., 2018) is trained on the Cityscapes dataset using a crop size of $768 \times 768$, a batch size of 8, and a total of 80K iterations. We employ stochastic gradient descent (SGD) with a momentum of 0.9 and a weight decay of 5e-4. The initial learning rate is set at 0.01. During training, we adjust the learning rate using the common 'poly' learning rate policy, which reduces the initial learning rate by multiplying $(1 - \frac{iter}{max\_iter})^{0.9}$. We apply standard data augmentation techniques, including random horizontal flipping and random resizing within the range of 0.5 to 2. For InternImage (Wang et al., 2023) on ADE20K, we use the same training settings in the paper (Wang et al., 2023). For PointRend (Kirillov et al., 2020) and Mask2Former (Cheng et al., 2022) on the LIS dataset, we adopt the same training settings as described in the paper (Chen et al., 2023).

## 4.3 ABLATION STUDY

Here, we conduct experiments to verify the effectiveness of the proposed De-aliasing filter(DAF) and frequency mixing module (FreqMix). We adopt widely used UPerNet (Xiao et al., 2018) with ResNet-50 (He et al., 2016) as the segmentation model owing to its effectiveness and simplicity.

Table 4: Comparison with blur filters on Cityscapes validation set.

| Method | mIoU↑ | BIoU↑ | BAcc↑ | FErr↓ | MErr↓ | DErr↓ | #FLOPs | #Params |
|---|---|---|---|---|---|---|---|---|
| UPerNet-R50 (Xiao et al., 2018) | 78.1 | 61.8 | 74.4 | 27.2 | 25.1 | 26.9 | 297.96G | 31.16M |
| Blur (Zhang, 2019) | 78.8 | 62.3 | 75.0 | 26.6 | 24.7 | 26.5 | 298.49G | 31.16M |
| AdaBlur (Zou et al., 2020) | 78.9 | 62.3 | 75.1 | 26.4 | 24.7 | 26.4 | 336.56G | 32.32M |
| FLC (Grabinski et al., 2022) | 78.6 | 62.7 | 74.9 | 26.7 | 24.8 | 26.7 | 297.96G | 31.16M |
| Ours (DAF) | 79.3 | 62.6 | 75.7 | 26.0 | 24.0 | 25.9 | 297.96G | 31.16M |
| Ours (DAF + FreqMix w/o channel-wise) | 79.4 | 62.8 | 75.9 | 25.5 | 23.8 | 25.5 | 298.54G | 31.22M |
| Ours (DAF + FreqMix) | **79.7** | **63.2** | **76.2** | **25.2** | **23.3** | **24.8** | 298.67G | 31.54M |

**Low-pass cut-off frequency of DAF.** Previous work (Grabinski et al., 2022) determines the low-pass cut-off frequency simply by the size of the down-sampling layer, *i.e.*, $\frac{1}{4}$. As shown in Table 3. By reducing all frequencies above $\frac{1}{4}$ to prevent aliasing, it improves the vanilla UPerNet by 0.5 mIoU and reduces displacement errors, false responses, and merging mistakes by 0.5, 0.3, and 0.2, respectively. By calculating the actual sampling rate with equation 7, we estimate a more accurate equivalent Nyquist frequency of $\frac{\sqrt{2}}{4}$, which achieves the best overall results (mIoU +1.2, BIoU +0.8) among settings where the low-pass cut-off frequency ranged from $\frac{1}{4}$ to $\frac{1}{4} \times 1.6$.

Table 5: Results on the PASCAL VOC dataset.

| Method | mIoU↑ | Boundary | | Three type errors | | |
|---|---|---|---|---|---|---|
| | | BIoU↑ | BAcc↑ | FErr↓ | MErr↓ | DErr↓ |
| UPerNet | 74.3 | 61.0 | 73.4 | 27.8 | 24.9 | 24.5 |
| Ours (+ DAF + FreqMix) | **76.1** | **62.7** | **74.9** | **26.0** | **24.0** | **23.6** |

**Effectiveness of FreqMix.** As shown in Table 4, with spatial-wise weighting, it improves the overall performance. Further weighting the frequency for the channel-wise component leads to an additional improvement, totaling +0.4 mIoU while reducing all three types of errors. Meanwhile, it only adds a minimal increase in FLOPs and parameters.

Table 6: Results on the ADE20K dataset.

| Method | mIoU↑ | Boundary | | Three type errors | | |
|---|---|---|---|---|---|---|
| | | BIoU↑ | BAcc↑ | FErr↓ | MErr↓ | DErr↓ |
| UPerNet | 38.9 | 29.5 | 40.5 | 60.4 | 58.9 | 60.2 |
| Ours (+ DAF + FreqMix) | **40.4** | **31.2** | **43.3** | **57.9** | **55.6** | **56.7** |

## 4.4 SEMANTICE SEGMENTATION

**Comparison with state-of-the-art blur filters.** As shown in Table 4, we compare the proposed DAF with former state-of-the-art blur filters designed to alleviate aliasing, including Blur (Zhang, 2019), AdaBlur (Zou et al., 2020), and FLC (Grabinski et al., 2022). Due to a more precise calculation of the actual sampling rate and the effective removal of the frequency power that leads to aliasing, the proposed DAF achieves the best results. Furthermore, when combined with FreqMix,

Table 7: Quantitative comparisons for low-light instance segmentation are conducted on the LIS test set (Chen et al., 2023), utilizing PointRend (Kirillov et al., 2020) and Mask2Former (Cheng et al., 2022) as instance segmentation models, with ResNet-50-FPN (He et al., 2016; Lin et al., 2017a) as the backbone. The training and testing settings align with those outlined in (Chen et al., 2023).

| Pipeline | Preprocessing method | Method | $AP^{mask}$ | $AP^{box}$ |
|---|---|---|---|---|
| Direct | - | PointRend | 20.6 | 23.5 |
| Enhance + Denoise | EnlightenGAN (Jiang et al., 2021) + SGN (Gu et al., 2019) | PointRend | 26.9 | 31.2 |
| | Zero-DCE (Guo et al., 2020) + SGN (Gu et al., 2019) | PointRend | 27.7 | 31.9 |
| Integrated | SID (Chen et al., 2018) | PointRend | 28.3 | 31.6 |
| Enhance + Denoise | REDI (Lamba & Mitra, 2021) | PointRend | 24.0 | 27.7 |
| End-to-end (Chen et al., 2023) | - | PointRend | 32.8 | 37.1 |
| End-to-end (**Ours**) | - | PointRend | **34.0 (+1.2)** | **38.2 (+1.1)** |
| Direct | - | Mask2Former | 21.4 | 22.9 |
| Enhance + Denoise | EnlightenGAN (Jiang et al., 2021) + SGN (Gu et al., 2019) | Mask2Former | 28.0 | 30.9 |
| | Zero-DCE (Guo et al., 2020) + SGN (Gu et al., 2019) | Mask2Former | 29.3 | 31.9 |
| Integrated | SID (Chen et al., 2018) | Mask2Former | 31.7 | 33.2 |
| Enhance + Denoise | REDI (Lamba & Mitra, 2021) | Mask2Former | 26.7 | 28.1 |
| End-to-end(Chen et al., 2023) | - | Mask2Former | 35.6 | 37.8 |
| End-to-end (**Ours**) | - | Mask2Former | **36.7 (+1.1)** | **38.8 (+1.0)** |

it leads to further overall improvements, outperforming the best competitor, AdaBlur, by 0.8 mIoU, and achieving the lowest error rates for all three types of hard pixels.

**Comparison across different challenging datasets.** As shown in Tables 5 and 6, we present a comparison of segmentation results across two challenging datasets: PASCAL VOC and ADE20K. They display our proposed method, denoted as 'Ours (+ DAF + FreqMix),' outperforming the UPerNet baseline across all metrics. Particularly, our method achieves a substantial improvement in mIoU, BIoU, and BAcc (increases of 1.8 mIoU, 1.7 BIoU, and 1.5 BAcc on PASCAL VOC, and 1.5 mIoU, 1.7 BIoU, and 2.8 BAcc on ADE20K). Additionally, the three types of errors are notably reduced (decreases of 1.8 FErr, 0.9 MErr, and 0.9 DErr on PASCAL VOC, and 2.5 FErr, 2.3 MErr, and 3.5 DErr on ADE20K), demonstrating the effectiveness of our approach in addressing hard pixels.

**Visualized results.** In Figure 7, our proposed method considerably enhances semantic segmentation quality, resulting in a substantial reduction in all three error types, including false responses, merging errors, and displacements. These are consistent with quantitative results.

### 4.5 Low-light instance Segmentation

We assess the effectiveness of our proposed method for the challenging task of low-light instance segmentation. The presence of noise in low-light images results in a high aliasing level, which can severely impair performance. As demonstrated in Table 7, our proposed method exhibits a substantial performance improvement (+1.2 AP for PointRend, +1.1 AP for Mask2Former) when compared to the previously established state-of-the-art pipelines described in (Chen et al., 2023).

## 5 Conclusion

In this study, we analyze three types of hard pixels that occur at object boundaries and establish a quantitative link between them and aliasing degradation. Specifically, we propose a method to calculate equivalent sampling rates for estimating the Nyquist frequency and aliasing levels during specific downsampling operations, employing a kernel larger than the stride and involving channel expansion. Moreover, we demonstrate the existence of a positive correlation between pixel-wise segmentation errors and the newly introduced aliasing score. Remarkably, these three types of challenging pixels exhibit distinct and unique patterns within the aliasing score, hinting at potential solutions tailored to scenarios that are sensitive to different types of errors.

These analyses empower us to precisely adjust the aliasing frequencies above the Nyquist frequency for specific downsampling operations. In this regard, we present two concrete solutions. The first solution, the De-Aliasing Filter (DAF), focuses on the modification of the downsampling layer, precisely removing frequencies that lead to aliasing. The second solution, the Frequency Mixing (FreqMix) module, involves a holistic adaptation of the encoder block, aiming to adjust the frequency distribution during feature extraction. Experimental results effectively validate the improved performance in both semantic segmentation and low-light instance segmentation tasks.

The insights gained from this analysis and the measurement tools we have introduced provide a fresh perspective and hold the potential for unlocking numerous research opportunities in this field.

ACKNOWLEDGMENTS

This work was supported by the National Natural Science Foundation of China (62088101 and 62171038), and JST Moonshot R&D Grant Number JPMJMS2011, Japan.

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

APPENDIX

This supplementary material provides more details and results that are not included in the main paper due to space limitations. The contents are organized as follows:

- Section A provides more analysis of aliasing degradation.
- Section B compares the aliasing score with confidence-based hard pixel type identification.
- Section C discusses the equivalent sampling rate when the kernel size and stride differ in height and width dimensions.
- Section D provides more implementation details of training.
- Section E includes an additional ablation study for the Frequency Mixing (FreqMix) module.
- Section F provides visualization of the frequency response of existing blur filters and demonstrates the advantage of the proposed de-aliasing filter.
- Section G provides additional experimental results combined with segmentation boundary refinement methods.
- Section H provides an analysis of the orthogonality of downsampling filters.
- Section I provides a detailed visual analysis, illustrating how aliasing degrades features and leads to three types of errors, and how DAF and FreqMix effectively address aliasing degradation.
- Section K analyzes the feature map in the frequency domain.
- Section L discusses the aliasing for the transformer-based architecture.

## A    ALIASING DEGRADATION

We have chosen ResNet (He et al., 2016), Swin Transformer (Liu et al., 2021b), and ConvNeXt (Liu et al., 2022) to perform a quantitative analysis of the correlation between aliasing scores and errors. Our analysis has been concentrated on the results obtained at object boundaries, where the majority of challenging pixels are found, as mentioned in (Li et al., 2017; Gu et al., 2020). ResNet is a well-established and widely used backbone, whereas Swin Transformer and ConvNeXt represent the latest transformer-based and CNN-based backbones, respectively. Despite the differences in their model structures, our findings, illustrated in Figure 8, reveal a consistent pattern: boundary pixels with higher aliasing scores tend to demonstrate higher cross-entropy errors, indicating that they are more prone to misclassification. These results highlight the pervasive issue of aliasing-induced degradation within modern deep neural networks. This observation underscores the need for comprehensive solutions to mitigate and address this problem, especially in the context of computer vision and other applications where accurate pixel-level information is crucial.

## B    COMPARING WITH THE PREDICTION CONFIDENCE METHOD

Previous research efforts (Li et al., 2017; Gu et al., 2020) have identified hard pixels based on prediction confidence. Here, we compare the prediction confidence method with the aliasing-based method for hard pixel identification. As illustrated in Figure 9, aliasing scores exhibit a more consistent trend than confidence-based results, especially when considering the effectiveness of distinguishing three types of errors: displacement errors, false responses, and merging mistakes. This finding underscores the potential of aliasing for categorizing errors effectively.

Notably, these errors display distinct characteristics when analyzed in the context of aliasing, and their importance varies across different scenarios. For instance, displacement errors tend to cluster in regions with high aliasing scores. In critical applications such as robotic surgery or radiation therapy, even a slight displacement of just two or three pixels from vital organs like the brainstem or the main artery can result in catastrophic consequences. Conversely, false responses and merging mistakes, which carry greater significance in autonomous driving scenarios, are more commonly found in areas with relatively low aliasing scores.

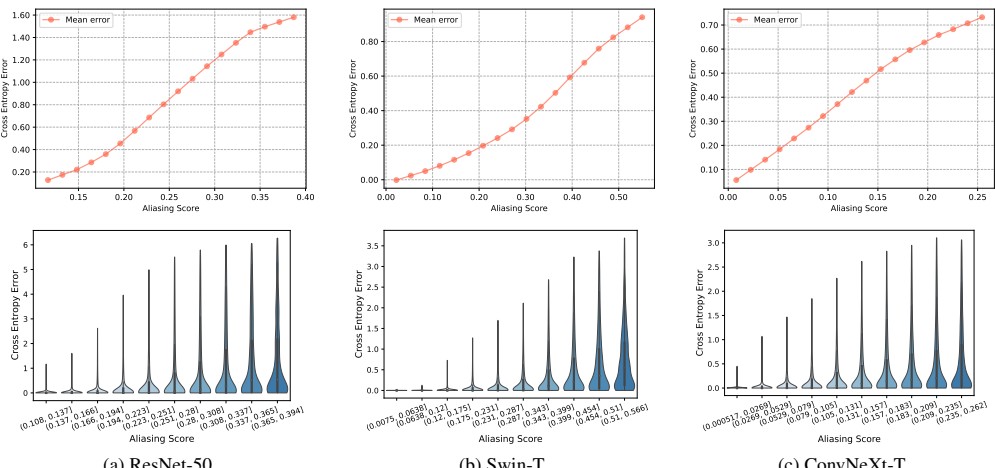

(a) ResNet-50      (b) Swin-T      (c) ConvNeXt-T

Figure 8: Illustration depicting the relationship between aliasing scores and hard pixels at boundaries. These results demonstrate the degradation caused by aliasing affecting three different widely-used models.

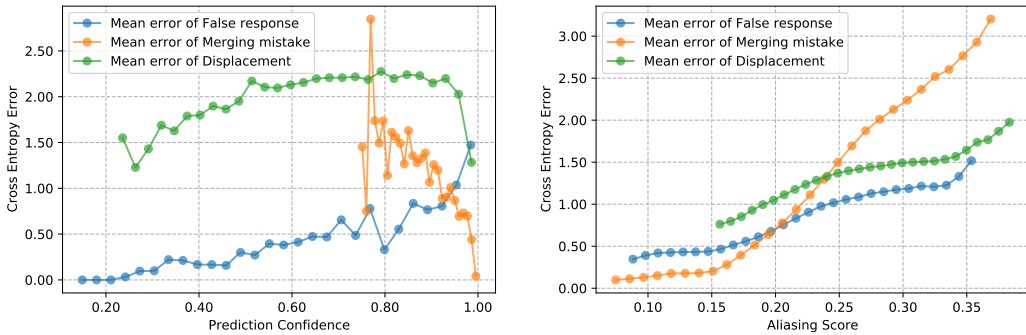

Figure 9: Left: Statistically correlated curve of cross-entropy error for three types of hard pixels with respect to prediction confidence (Li et al., 2017). Right: Distribution of the three types of hard pixels.

This analysis not only provides insights into distinguishing between the three error types but also offers valuable guidance for designing error-correcting methods tailored to specific scenarios.

## C   EQUIVALENT SAMPLING RATE

In the main paper, we provide the calculation of the equivalent sampling rate when the downsampling kernel and stride are the same in the height and width dimensions, which is the most common situation in existing modern deep neural networks (He et al., 2016; Liu et al., 2021b; 2022; Wang et al., 2023). Here, we discuss the equivalent sampling rate when the kernel size and stride in the height and width dimensions are different.

Similarly, we consider both the kernel size and feature size (channel, height, and width), rather than just the downsampling stride. We introduce a simple equation for the equivalent sampling rate in the height and width dimensions ($\text{ESR}^H$, $\text{ESR}^W$) to calculate the actual sampling rate as follows:

$$
\begin{aligned}
\text{ESR}^H &= \min(K_{\text{down}}^H, \sqrt{\frac{C^{\text{out}}}{C^{\text{in}}}}) \times \frac{H^{\text{out}}}{H^{\text{in}}}, \\
\text{ESR}^W &= \min(K_{\text{down}}^W, \sqrt{\frac{C^{\text{out}}}{C^{\text{in}}}}) \times \frac{W^{\text{out}}}{W^{\text{in}}},
\end{aligned}
\tag{7}
$$

where $C$, $H$, and $W$ are the size of the channel, height, and width. "in" and "out" indicate the input and output features. $K_{\text{down}}^H$, $K_{\text{down}}^W$ present the downsampling kernel size in height and width dimensions. $\frac{H^{\text{out}}}{H^{\text{in}}}$, $\frac{W^{\text{out}}}{W^{\text{in}}}$ are equal to downsampling stride in height and width dimension, which

Table 8: Ablation study for Frequency Mixing (FreqMix) module.

| Method | mIoU↑ | BIoU↑ | BAcc↑ | FErr↓ | MErr↓ | DErr↓ | #FLOPs | #Params |
|---|---|---|---|---|---|---|---|---|
| FreqMix w/ decomposition | 79.6 | 58.6 | 74.7 | **23.5** | 52.9 | 26.3 | 423.69G | 39.64M |
| FreqMix w/o decomposition | **79.7** | **58.8** | **74.9** | 24.0 | **52.8** | **26.1** | **298.67G** | **31.54M** |

aligns with (Grabinski et al., 2022). $\min(K_{\text{down}}^H, \sqrt{\frac{C^{\text{out}}}{C^{\text{in}}}})$, $\min(K_{\text{down}}^W, \sqrt{\frac{C^{\text{out}}}{C^{\text{in}}}})$ indicates the influence of the downsampling kernel size $K^{\text{down}}$ and channel expansion. Notice that we make the common assumption that the impact of channel expansion works for both height and width dimensions, using the square root to calculate the impact for both dimensions.

## D  MORE IMPLEMENT DETAILS

For the Cityscapes dataset, we employ a crop size of $768 \times 768$, a batch size of 8, and a total of 80K iterations. For the PASCAL VOC dataset, we utilize a crop size of $512 \times 512$, a batch size of 16, and a total of 40K iterations. On the ADE20K dataset, we adopt a crop size of $512 \times 512$, a batch size of 16, and a total of 80K iterations. We also set the channel of the feature pyramid to 128. We employ stochastic gradient descent (SGD) with a momentum of 0.9 and a weight decay of 5e-4. The initial learning rate is set at 0.01. During training, we adjust the learning rate using the common 'poly' learning rate policy, which reduces the initial learning rate by multiplying $(1 - \frac{\text{iter}}{\text{max\_iter}})^{0.9}$. We apply standard data augmentation techniques, including random horizontal flipping and random resizing within the range of 0.5 to 2.

For PointRend (Kirillov et al., 2020) and Mask2Former (Cheng et al., 2022) on the LIS dataset, we adopt the same training settings as described in the paper (Chen et al., 2023). We use random flip as data augmentation and train with a batch size of 8, a learning rate of 1e-2 for 12 epochs, with a learning rate dropping by $10\times$ at 8 and 11 epochs, respectively. To make the model quickly adapt to low-light settings, we use COCO pre-trained model as initialization following (Chen et al., 2023).

## E  ABLATION STUDY

In this section, we present an additional ablation study focused on the Frequency Mixing (FreqMix) module. In the main paper, we describe our approach to predicting weighting values. Specifically, we decompose the prediction of three-dimensional frequency weighting values $A^\downarrow, A^\uparrow \in \mathbb{R}^{C \times H \times W}$, into channel-wise components $A_1^\downarrow, A_2^\uparrow \in \mathbb{R}^{H \times W}$ and spatial-wise components $A_1^\downarrow, A_2^\uparrow \in \mathbb{R}^{H \times W}$. Where $A^\downarrow$ represents the weighting values for frequencies below the Nyquist frequency, and $A^\uparrow$ represents those above Nyquist frequency. As shown in Table 8, the prediction of three-dimensional frequency weighting values can be computationally intensive for prediction. It directly introduces an additional 7.2 million parameters and requires an extra 125.1 GFLOPs of computational cost. Interestingly, despite this increase in complexity, the overall results are largely similar to the decomposition solution. This highlights the efficiency of our proposed method.

## F  VISUALIZATION OF VARIOUS BLUR FILTERS

For better comparison, we also illustrate the visualization of the frequency response of existing blur filters, including Blur (Zhang, 2019), AdaBlur (Zou et al., 2020), FLC (Grabinski et al., 2022), and our proposed methods in Figure 10. The top row shows the frequency response in two dimensions. We shift the low frequency to the center, and the four corners indicate high frequency, with brighter areas representing higher response. The bottom row displays the frequency response in one dimension, where the left side represents lower frequency, and the right side represents higher frequency. The red line indicates the Nyquist frequency. We observe that Blur (Zhang, 2019) and AdaBlur (Zou et al., 2020) cannot entirely eliminate frequencies higher than the Nyquist frequency. Conversely, due to an underestimation of the Nyquist frequency, FLC (Grabinski et al., 2022) excessively removes frequencies below the Nyquist frequency, resulting in information loss. In contrast, our proposed de-aliasing filter effectively and precisely removes the frequency power above the Nyquist frequency, which explains its effectiveness.

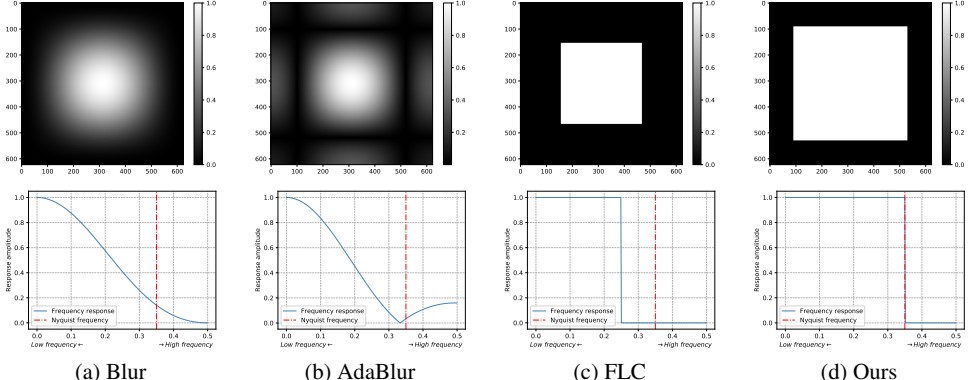

| (a) Blur | (b) AdaBlur | (c) FLC | (d) Ours |

Figure 10: Visualization of the frequency response of existing blur filters, including Blur (Zhang, 2019), AdaBlur (Zou et al., 2020), FLC (Grabinski et al., 2022), and our proposed methods. The top row shows the frequency response in two dimensions. We shift the low frequency to the center, and the four corners indicate high frequency, with brighter areas representing higher response. The bottom row displays the frequency response in one dimension, where the left side represents lower frequency, and the right side represents higher frequency. The red line indicates the Nyquist frequency.

Table 9: Combination with boundary refinement methods on the Cityscapes (Cordts et al., 2016) validation set. Results are reported from the original paper (Yuan et al., 2020).

| Method | DeepLabv3 [arXiv2017] (Chen et al., 2017) | +GUM [BMVC2018] (Mazzini, 2018) | +DenseCRF [NeurIPS] (Krähenbühl & Koltun, 2011) | +SegFix [ECCV2020] (Yuan et al., 2020) | +SegFix+Ours (Ours) |
|---|---|---|---|---|---|
| mIoU | 79.5 | 79.8 | 79.7 | 80.5 | **81.1** |

Table 10: Orthogonality analysis for downsampling filters in ResNet (He et al., 2016), Swin Transformer (Liu et al., 2021b), ConvNeXt (Liu et al., 2022), HorNet (Rao et al., 2022), and DiNAT (Hassani & Shi, 2022). Their weights are obtained by training on ImageNet. A higher absolute cosine similarity value indicates greater similarity in filter weights, suggesting a lower degree of orthogonality (0.0 = totally orthogonal, 1.0 = identical filters).

| Model | ResNet-18 | ConvNeXt-T | Swin-T | HorNet-T | DiNAT-L |
|---|---|---|---|---|---|
| Abs. CosSim. | 0.067 | 0.072 | 0.06 | 0.069 | 0.046 |

## G  COMBINATION WITH BOUNDARY REFINEMENT METHODS

In this section, we integrate our proposed method with SegFix (Yuan et al., 2020), a previously effective approach for semantic segmentation boundary refinement. SegFix significantly improves segmentation results by refining predictions, particularly at boundaries where most hard pixels occur. Our proposed method operates independently from SegFix, enhancing models by optimizing intermediate features, precisely removing frequencies leading to aliasing (via the de-aliasing filter), and adjusting frequencies using the encoder block (Frequency Mixing Module).

The synergy between these techniques is evident in the results presented in Table 9, where our method enhances SegFix by an additional 0.6 mIoU. This improvement highlights the effectiveness of our approach in addressing complex segmentation challenges.

## H  DOWNSAMPLING FILTERS ORTHOGONALITY ANALYSIS

The introduced calculation of the equivalent sampling rate is based on the assumption that downsampling filters are orthogonal. In this section, we quantitatively analyze the orthogonality degree of downsampling filters in widely used models, including ResNet (He et al., 2016), Swin Transformer (Liu et al., 2021b), ConvNeXt (Liu et al., 2022), HorNet (Rao et al., 2022), and DiNAT (Hassani & Shi, 2022). Their weights are obtained by training on ImageNet.

As shown in Table 10 and Figure 11, we use the absolute cosine similarity as the quantitative measurement of the orthogonality degree. A higher absolute cosine similarity value indicates greater

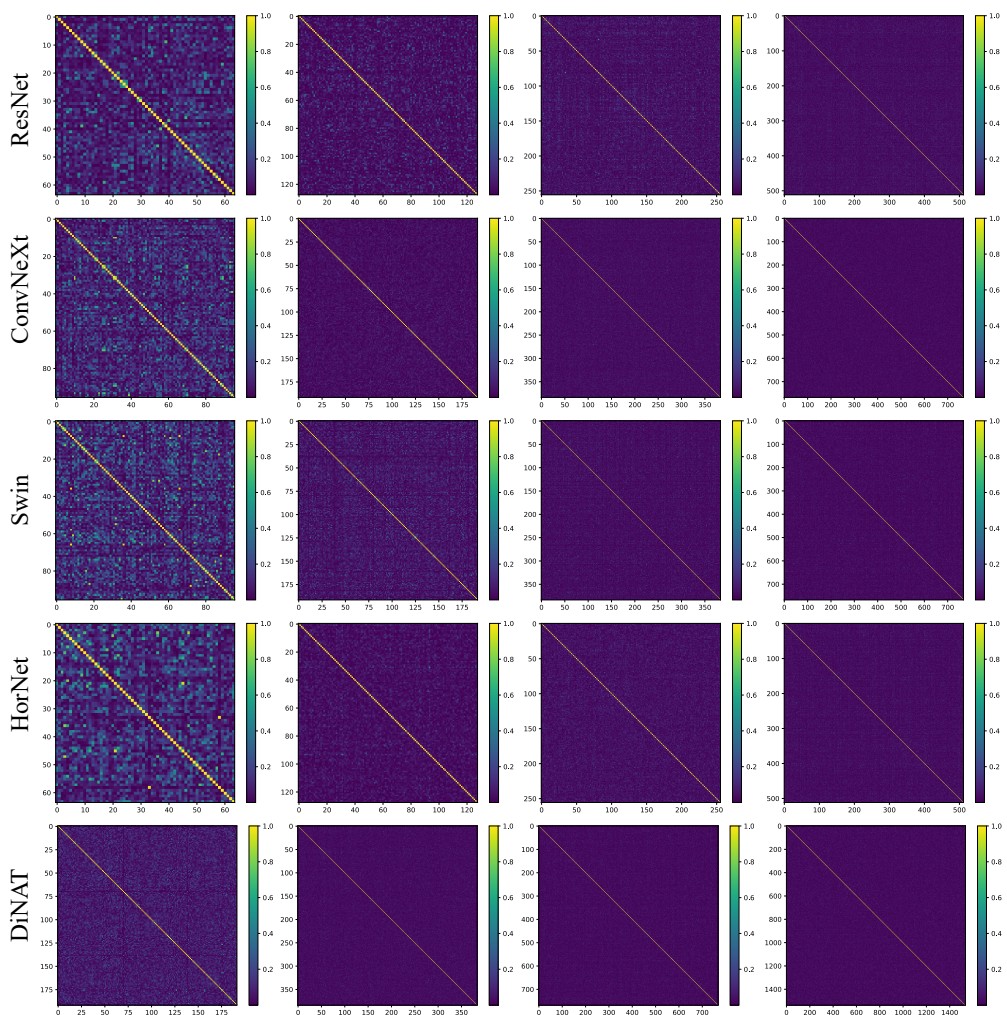

Figure 11: Orthogonality degree analysis of filters for downsampling in ResNet (He et al., 2016), Swin Transformer (Liu et al., 2021b), ConvNeXtcitep2022convnet, HorNet(Rao et al., 2022), and DiNAT (Hassani & Shi, 2022). We illustrate the absolute cosine similarity matrix, where each element indicates the absolute cosine similarity between different filters. A brighter color indicates a higher similarity in filter weights, suggesting a lower degree of orthogonality. We observe the matrix showing dark colors, with bright colors only appearing along the diagonal (self-to-self), indicating that the filters are essentially orthogonal.

similarity in filter weights, suggesting a lower degree of orthogonality. As depicted in Table 10 and Figure 11, the quantitative measurements indicate that the downsampling filters are predominantly orthogonal (with an average absolute cosine similarity ranging from 0.046 to 0.072), thereby supporting the introduced equivalent sampling rate in Section C. The proposed equivalent sampling rate is designed as a heuristic for selecting the cutoff frequency, and we think that exploring how to finely adjust the equivalent sampling rate based on the orthogonality of the filter is a very interesting and important problem that is worth further investigation.

# I   VISUALIZED ANALYSIS FOR DAF AND FREQMIX

To investigate how the proposed DAF and FreqMix address aliasing degradation and enhance deep neural networks, we visualize the deep features in the model and randomly select some examples in Figures 12 and 13. Furthermore, in Figures 14, we visualize three types of errors and demonstrate how the proposed DAF and FreqMix alleviate these errors: 1) false responses, 2) merging mistakes, and 3) displacements.

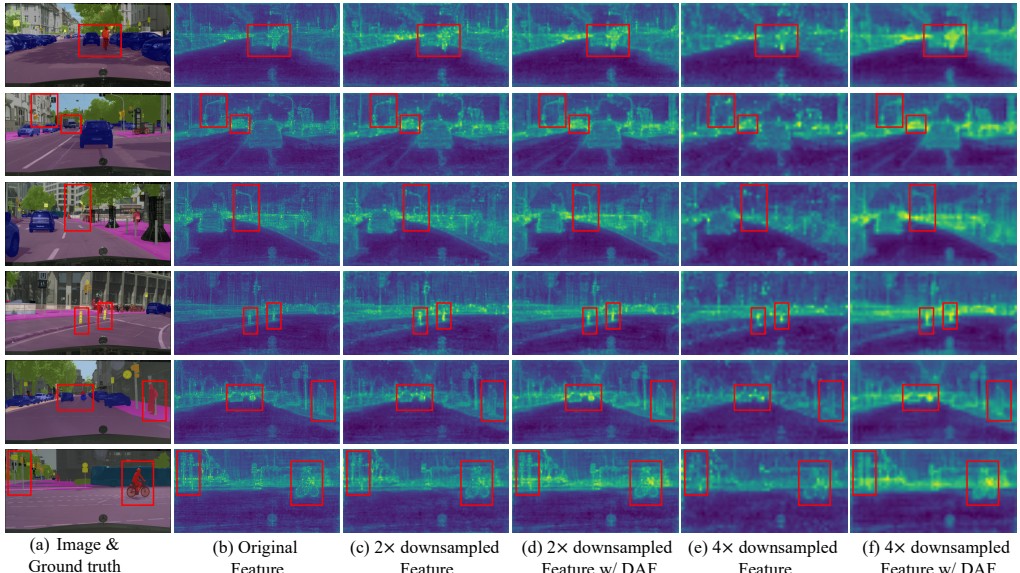

| (a) Image & Ground truth | (b) Original Feature | (c) 2× downsampled Feature | (d) 2× downsampled Feature w/ DAF | (e) 4× downsampled Feature | (f) 4× downsampled Feature w/ DAF |

Figure 12: Visualization for DAF. We mark some high aliasing score areas in the feature with a red box. Without DAF in (c) and (e), the downsampling of features exhibits a severe "jagged" phenomenon (Zou et al., 2020; Qian et al., 2021), resulting in the degraded representation of object boundaries. The response of some objects is faded or lost in the 4× scale in (e). By directly removing the high frequency leads to aliasing degradation, the proposed DAF can largely relieve the "jagged" phenomenon (Zou et al., 2020; Qian et al., 2021), making the boundaries more clear in the (d) and (f). Furthermore, DAF largely preserves the object responses, as shown in (f), compared to (e).

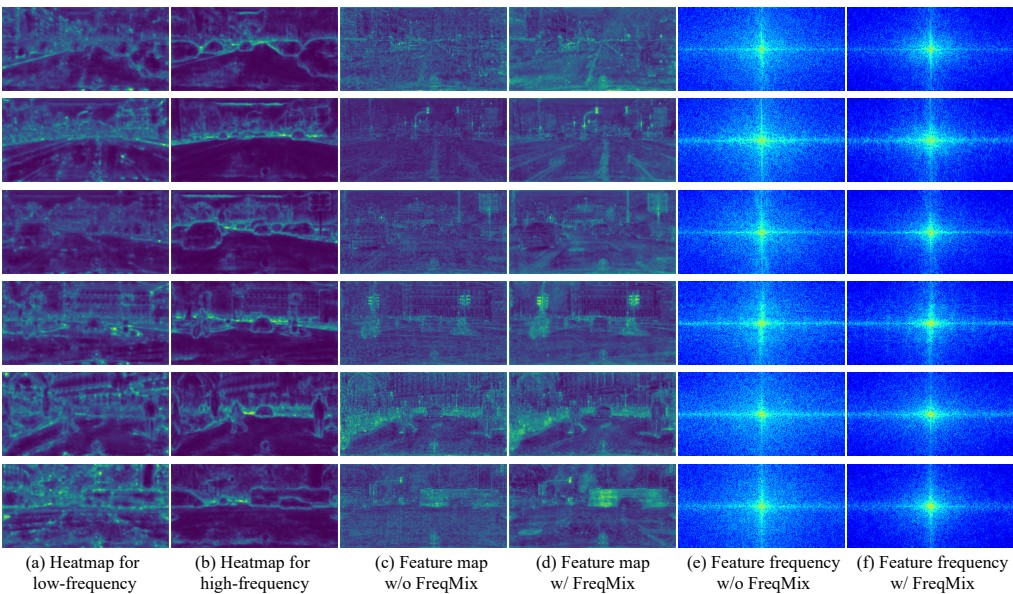

| (a) Heatmap for low-frequency | (b) Heatmap for high-frequency | (c) Feature map w/o FreqMix | (d) Feature map w/ FreqMix | (e) Feature frequency w/o FreqMix | (f) Feature frequency w/ FreqMix |

Figure 13: Visualization for FreqMix. In (a) and (b), the heatmap shows a brighter color for object boundaries and a darker color for object centers and backgrounds, especially for high-frequency components, where a brighter color indicates a high value. Thus, FreqMix not only reduces the overall high-frequency content responsible for aliasing degradation in (f) but also preserves the high frequency of object boundaries. This preservation is crucial for making the boundaries clear in (d), ensuring accurate segmentation, and lowering the occurrence of three types of errors.

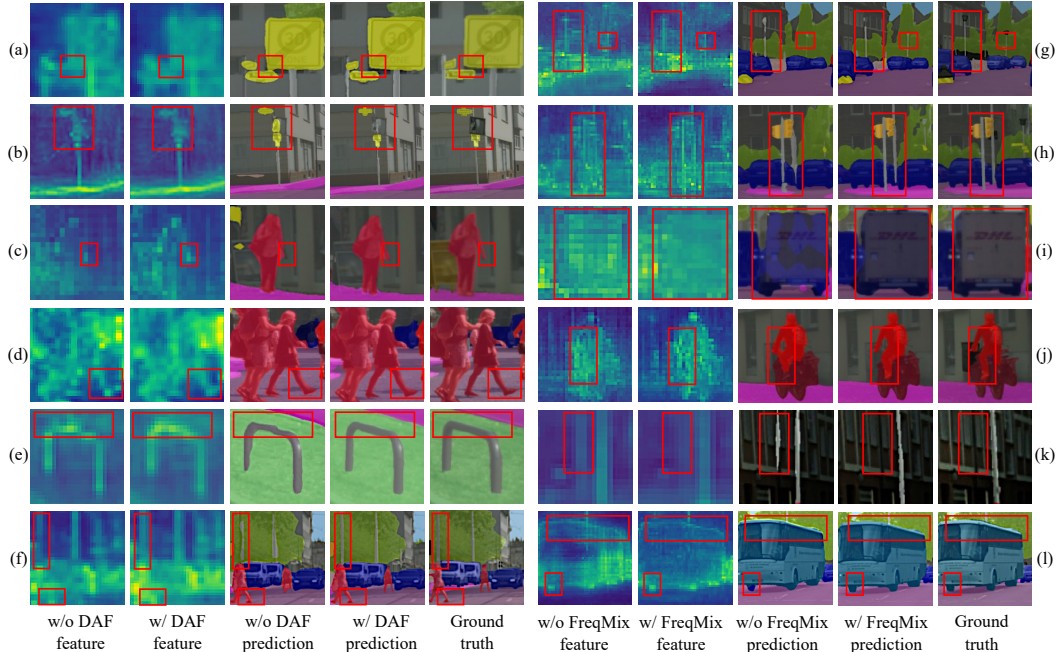

w/o DAF feature    w/ DAF feature    w/o DAF prediction    w/ DAF prediction    Ground truth    w/o FreqMix feature    w/ FreqMix feature    w/o FreqMix prediction    w/ FreqMix prediction    Ground truth

Figure 14: Visualization of how the proposed DAF and FreqMix address aliasing degradation and improve the feature and final segmentation. We randomly select image patches with a high aliasing score from Cityscapes (Cordts et al., 2016) dataset validation set. Zoom in for better view.

## I.1 VISUALIZATION FOR DE-ALIASING FILTER (DAF)

As depicted in Figures 12(c) and (e), we indicate some areas with a high aliasing score in the feature with the red box, they exhibit a severe "jagged" phenomenon (Zou et al., 2020; Qian et al., 2021), leading to the degraded representation of object boundaries. Moreover, in comparison with the original feature in Figure 12(b), the response of some objects is lost in the 4× downsampling in Figures 12(e). It is noteworthy that widely used state-of-the-art models, such as Swin Transformer (Liu et al., 2021b) and ConvNeXt (Liu et al., 2022), adopt a total downsampling stride of 32×, potentially leading to an even more severe loss of response.

By directly removing the high frequency leads to aliasing degradation, the proposed DAF can largely relieve the "jagged" phenomenon (Zou et al., 2020; Qian et al., 2021), making the boundaries more clear in the Figures 12(d) and (f). Furthermore, DAF largely preserves the object responses, as shown in Figure 12(f), compared to Figure 12(e), resulting in a more accurate segmentation prediction and a lower occurrence of the three types of errors shown in the left column Figure 14.

## I.2 VISUALIZATION FOR FREQUENCY MIXING MODULE (FREQMIX)

FreqMix improves the model by decomposing features into low frequency and high frequency using a Nyquist frequency threshold and dynamically selecting them in a spatial-variant manner. We visualize the heatmap for selecting low frequency and high frequency in Figures 13(a) and (b), where a brighter color indicates a high value. The heatmap shows a brighter color for object boundaries and a darker color for object centers and backgrounds, especially for high-frequency components. Thus, FreqMix not only reduces the overall high-frequency content (see Figure 13(e) and (f)), responsible for aliasing degradation, but also preserves the high frequency of object boundaries. This preservation is crucial for making the boundaries clear (see Figure 13(c) and (d)), ensuring accurate segmentation and a lower occurrence of three types of errors. Further visualization in the right column Figure 14 verifies that FreqMix reduces the occurrence of the three types of errors.

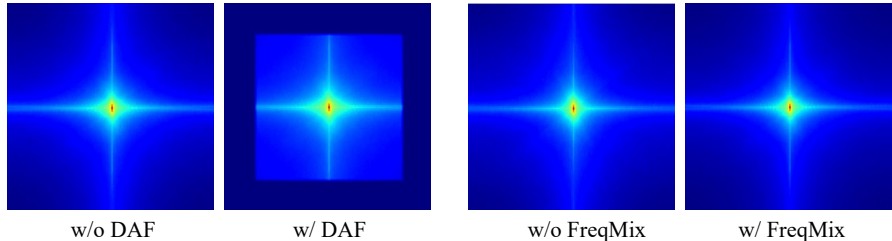

| w/o DAF | w/ DAF | w/o FreqMix | w/ FreqMix |

Figure 15: Visualization of the averaged frequency distribution of features. The center indicates low frequency, and the corners indicate high frequency. A brighter color indicates more corresponding frequency components. The DAF directly removes the frequency above the Nyquist frequency, while FreqMix suppresses the high frequency.

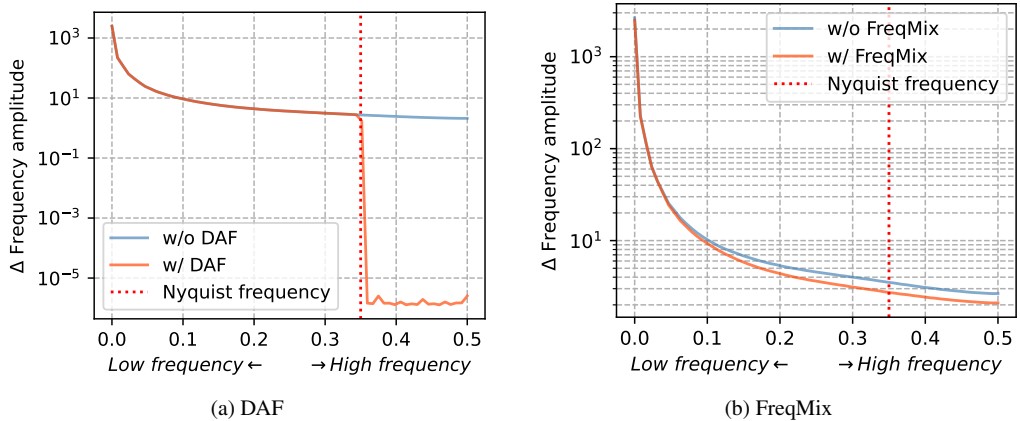

Figure 16: Visualization of the frequency distribution of features.

## I.3 VISUALIZATION OF THREE TYPES OF ERRORS

As illustrated in Figure 14, we randomly selected image patches with a high aliasing score from the Cityscapes (Cordts et al., 2016) dataset validation set.

We observed that aliasing leads to a "jagged" phenomenon (Zou et al., 2020; Qian et al., 2021), disrupting object shapes and boundaries, resulting in false responses (Figures 14(b) and (l)) and displacement errors (Figures 14(c), (d), and (e)). DAF and FreqMix relieve this phenomenon and improve the feature representation thus resulting in lower false responses and displacement errors.

When high-frequency information is aliased, it transforms into false low-frequency information. For example, when two objects are close to each other, their high-frequency boundaries can be aliased to lower frequency during downsampling, causing the two objects to appear connected and their boundaries to be merged. This leads to merging errors (Figures 14(a), (g), and (j)). DAF/FreqMix solve this by removing/suppressing these high frequencies during downsampling/encoder block, leading to lower merging errors.

Moreover, high-frequency components in the object center or background can result in false responses (Figures 14(i) and (k)). FreqMix addresses this issue by suppressing the high frequency in the object center or background while preserving the high frequency at the boundaries.

## J FEATURE FREQUENCY ANALYSIS

We present a feature frequency analysis in Figures 15 and 16. The DAF directly eliminates frequencies above the Nyquist frequency, while the FreqMix suppresses high-frequency components, alleviating aliasing degradation.

## K  FEATURE FREQUENCY ANALYSIS

We present a feature frequency analysis in Figures 15 and 16. The DAF directly eliminates frequencies above the Nyquist frequency, while the FreqMix suppresses high-frequency components, alleviating aliasing degradation.

## L  DISCUSSION ABOUT ALIASING IN A TRANSFORMER-BASED ARCHITECTURE

As for recent transformer-based architectures, aliasing remains a concern. Taking the renowned Vision Transformer (ViT) as an example, ViT (Dosovitskiy et al., 2020) tokenizes images by splitting them into non-overlapping patches, which are then fed into transformer blocks. The tokenization and self-attention operations performed on these discontinuous patch embeddings can be viewed as downsampling operations, introducing a potential side effect of aliasing. It is essential to note that this downsampling operation is virtually unavoidable due to the spatial redundancy nature of the image (He et al., 2022) and huge computational costs without downsampling (increasing by $256\times$ without downsampling in ViT).

Several existing studies have acknowledged this concern. A straightforward solution to alleviate aliasing is to increase the sampling rate. Similar trends are observed in vision transformers, where the use of overlapped tokens (Yuan et al., 2021) and smaller patch sizes (Caron et al., 2021) contributes to improved performance. However, escalating sampling rates incur quadratic computational costs. Consequently, I hypothesize that integrating appropriate anti-aliasing filters into the 'attending' process could offer a viable solution. In fact, existing work has empirically explored blending anti-aliasing filters into the vision transformer, reporting observed improvements (Qian et al., 2021).

In conclusion, aliasing persists as a potential concern in transformer-based architectures, and prior studies have endeavored to address it empirically by enhancing the sampling rate or integrating anti-aliasing filters into the attention mechanism. Our work represents a step forward in quantitatively assessing and addressing aliasing in contemporary models, supported by theoretical foundations. This issue still presents ample opportunities for further investigation and exploration.

