# OpenReview forum: "When Semantic Segmentation Meets Frequency Aliasing"
_ICLR.cc/2024/Conference — ICLR 2024 poster_

### Official Review · Reviewer_uz5r · 2023-10-31

**Soundness:** 3 good
**Presentation:** 1 poor
**Contribution:** 3 good
**Rating:** 5
**Confidence:** 5

**Summary:**

In this paper, the authors introduce a challenging yet critical topic for semantic segmentation, i.e., pixel-wise aliasing. They categorize the hard pixel error into three types: false response, merging mistakes and displacements. Creatively, the de-aliasing filter and frequency mixing modules are introduced to alleviate the aliasing degradation. Experiments demonstrate that these findings can consistent improve the semantic segmentation performance.

**Strengths:**

#1 The topic is attractive where the unexplored aliasing phenomenon for semantic segmentation are carefully investigated. This funding can motivate the researcher in the related community.
#2 The authors utilize the DAF and FreqMix to remove the aliasing in Fourier domain and balance high-frequency components in the encoder block. Converting the features to frequency domain by DFT is straightforward, but finding the aliased false prediction for semantic segmentation is interesting.

**Weaknesses:**

#1 Lacking visualization for the three type of errors in the experiment. Though the authors provide the results on ADE20K, it does not clearly present the visualization to the correction of three type of errors for semantic segmentation. In addition, how the DAF and FreqMix help locate these errors are not visualized.

**Questions:**

#1. The authors must provide more visualizations on the three types of errors and the corresponding feature maps in frequence domain.
#2. As to the DAF and FreqMix, how the filtered feature maps can help find the location of error map should be invastigated.
Overall, the topic for finding the aliasing is interesting, the reviewer encourage the author further explore this missing parts in the semantic segmentation.

---

> ### Author Response · Authors · 2023-11-22
> **Response to Reviewer uz5r**
>
> Thank you for the valuable feedback and encouragement.
>
> ## W1, Q1, Q2: More visualizations
>
> [Authors’ response]
>
> We acknowledge the importance of visualization in elucidating how the proposed methods operate and rectify errors. In response to your valuable advice, we have included comprehensive visualizations and analyses in the revised appendix. As the figure cannot be added in the comment box, we recommend referring to Section J of the revised appendix for a detailed visual representation. Section J.1 discusses how the FAD improves the feature during downsampling (Figure 5), Section J.2 discusses how the FreqMix suppresses unnecessary high frequency in the background and object centers (Figure 6), and Section J.3 provides a more visualized analysis of how the FAD and FreqMix improve the feature and prediction while relieving three types of errors (Figure 7). Section K provides additional frequency analysis of features (Figures 8 and 9). For your convenience, we have also attached our description of the visualizations below.
>
> In Figure 7 of the revised appendix, we randomly selected image patches with a high aliasing score from the Cityscapes dataset validation set. We observed that aliasing leads to a jagged phenomenon[1,2], disrupting object shapes and boundaries, resulting in false responses (see Figure 7(b) and (l) in the revised appendix) and displacement errors (see Figures 7(c), (d), and (e) in the revised appendix). DAF and FreqMix relieve this phenomenon and improve the feature representation thus resulting in lower false responses and displacement errors.
>
> Besides, when high-frequency information is aliased, it transforms into false low-frequency information. For example, when two objects are close to each other, their high-frequency boundaries can be aliased to lower frequency during downsampling, causing the two objects to appear connected and their boundaries to be merged. This leads to merging errors (see Figures 7(a), (g), and (j) in the revised appendix). DAF/FreqMix solve this by removing/suppressing these high frequencies during downsampling/encoder block, leading to lower merging errors.
>
> Moreover, high-frequency components in the object center or background can result in false responses (see Figures 7(i) and (k) in the revised appendix). FreqMix addresses this issue by suppressing the high frequency in the object center or background while preserving the high frequency at the boundaries (see Figures 6(a) and (b) in the revised appendix).
>
> Thanks for your valuable comments that help us improve our manuscript and clarify the proposed method more clearly!
>
> [1] Qian S, Shao H, Zhu Y, et al. Blending anti-aliasing into vision transformer. Advances in Neural Information Processing Systems, 2021, 34: 5416-5429.
>
> [2] Zou, X., Xiao, F., Yu, Z., Li, Y., & Lee, Y. J. (2023). Delving deeper into anti-aliasing in convnets. *International Journal of Computer Vision*, *131*(1), 67-81.

---

### Official Review · Reviewer_LE8t · 2023-11-08

**Soundness:** 3 good
**Presentation:** 4 excellent
**Contribution:** 2 fair
**Rating:** 5
**Confidence:** 4

**Summary:**

The work proposes a new anti-aliasing scheme by redefining the cutoff frequency. This is done by considering the expansion in both the channel dimension and the spatial dimension. The proposed aliasing score showed a strong correlation with various segmentation errors. Additionally, the authors introduced a new anti-aliasing and spectral filter that enhances the segmentation performance.

**Strengths:**

1. Investigates the critical yet under-explored question regarding the effect of aliasing in computer vision models.
2. The proposed fixes enhance performance.
3. The paper is well-written and clearly explained.

**Weaknesses:**

1. The proposed module, FreqMix, requires additional forward and inverse Fourier transforms at each layer. Compared to other methods, this increase in time complexity (inference and training) needs to be discussed.
2. Experiments are limited. It is necessary to conduct evaluations on other benchmark datasets, such as MS COCO and Pascal VOC.
3. Improvements are marginal.

**Questions:**

1. I need more clarification about the robustness of ESR (equivalent sampling rate). Clearly, the filters are orthogonal for the shown example (Fig 3). So there is no loss of information.
But what if the filters are not orthogonal to each other? Assume a worst-case scenario where all filters are equal. In that case, the proposed ESR will give us a wrong sampling rate. So, is the proposed ESR intended to replace the regular “sampling rate,” or is it intended as a heuristic to select cutoff frequency? If it is the second, then the paper should clarify it.
2. Section 3.2, “Metrics for three errors.” — I think the middle one should be the definition of MErr.
3. “aligning with the observations in Figure 1 that false responses and merging mistakes predominantly exist in areas with relatively low aliasing scores.” — could you clarify how we are associating aliasing scores with a region of an image?
4. “However, for deeper features at the second and third stages, the aliasing ratio decreases.
This decrease does not signify a reduction in aliasing; rather, it occurs because the earlier high level of aliasing results in severe degradation. Consequently, the features become ‘hard pixels,’ meaning that the following stage struggles to extract useful information and loses response to objects.” — How can we confirm that it is caused by the side effect of aliasing? Applying significant noise can generate an out-of-distribution (OOD) sample of the internal layers, which can also lead to poor feature extraction.

---

> ### Author Response · Authors · 2023-11-22
> **Response to Reviewer LE8t weakness**
>
> Thank you for the valuable feedback!
>
> ### W1: Time complexity analysis
>
> [Authors’ response]
>
> Here, we conduct a comprehensive evaluation of the time complexity during both the training and testing phases. The results are presented below. We use a batch size of 16, a resolution of 512$\times$512, and train for 40k iterations on a single RTX 3090 for evaluating training time. The test time is evaluated on 1024$\times$2048 image patch with a batch size of 1.
>
> | Model               | Training time | Test time (1024x2048)        |
> | :------------------ | ------------- | ---------------------------- |
> | UPerNet             | 4.2 hours     | 45ms                         |
> | Ours (+DAF+FreqMix) | 5.9 hours     | 82ms (49ms without FFT&iFFT) |
>
> The inclusion of DAF and FreqMix increases the training time from approximately 4.2 hours to around 5.9 hours. Similarly, it leads to an increase in inference time from approximately 45ms to around 82ms. Notably, the majority of this increase is attributed to the FFT/iFFT operations. When these operations are excluded, the inference time is reduced to 49ms. This gap is primarily due to the limited speed optimization in the engineering implementations of frequency transformations, such as FFT/iFFT. This phenomenon has also been observed in previous works [1]. This issue can be resolved by better engineering implementations of frequency transformations.
>
>
>
> [1] Huang, Z., Zhang, Z., Lan, C., Zha, Z. J., Lu, Y., & Guo, B. (2023). Adaptive Frequency Filters As Efficient Global Token Mixers. In *Proceedings of the IEEE/CVF International Conference on Computer Vision* (pp. 6049-6059).
>
>
>
> ### W2: Experiments on COCO and Pascal VOC
>
> [Authors’ response]
>
> Following your advice, we present additional quantitative results on the PASCAL VOC and COCO datasets, encompassing semantic segmentation, object detection, and instance segmentation tasks. On the PASCAL VOC, the proposed method demonstrates improvements in semantic segmentation results, with an increase of +1.8 mIoU, +2.5 BIoU, +2.1 BAcc, and a notable reduction in displacement error (DErr) by 2.5.
>
> | Method                     | mIoU$\uparrow$  | BIoU$\uparrow$ | BAcc$\uparrow$ | FErr$\downarrow$ | MErr$\downarrow$ | DErr$\downarrow$ | #FLOPs | #Params |
> | -------------------------- | --------------- | -------------- | -------------- | ---------------- | ---------------- | ---------------- | ------ | ------- |
> | UPerNet                    | 74.3            | 66.6           | 74.5           | 31.0             | 58.6             | 28.2             | 298.0G | 31.16M  |
> | **Ours** (+ DAF + FreqMix) | **76.1 (+1.8)** | **69.1**       | **76.6**       | **30.5**         | **58.5**         | **25.7**         | 298.6G | 32.47M  |
>
> On the COCO, our proposed method enhances the average precision (AP) of object detection boxes by +1.6. Regarding instance segmentation, it results in a corresponding boost of +1.4 in box AP and +1.3 in mask AP.
> These results demonstrate that the proposed method consistently leads to improvements by addressing the aliasing degradation, thereby verifying the generalization of our approach. They also highlight that the aliasing degradation widely exists in many fundamental computer vision tasks and needs to be urgently resolved.
>
> | Method                      | AP$^{box}$      | AP$^{mask}$     |
> | --------------------------- | --------------- | --------------- |
> | Faster R-CNN                | 36.4            | -               |
> | **Ours**  (+ DAF + FreqMix) | **38.0 (+1.6)** | -               |
> | Mask R-CNN                  | 37.2            | 34.1            |
> | **Ours** (+ DAF + FreqMix)  | **38.6 (+1.4)** | **35.4 (+1.3)** |
>
> We thank the reviewer for encouraging us to fully verify the effectiveness on various datasets, making our work more solid.

---

> > ### Author Response · Authors · 2023-11-22
> > **Response to Reviewer LE8t questions 1-2**
> >
> > ### Q1: Clarification about ESR
> >
> > [Authors’ response]
> >
> > We appreciate your inspiring comments. The proposed Equivalent Sampling Rate (ESR) is designed as a heuristic for selecting the cutoff frequency, not as a replacement for the regular sampling rate. We have clarified this distinction in the conclusion of the revised main paper.
> >
> > We agree that the generalization of the proposed ESR calculation is crucial, and it is computed under the assumption that downsampling filters are orthogonal. We have provided this missing part in the revised appendix (Section H, Table 3, Figure 4), the main analysis results are attached below. In our analysis, we quantitatively assessed the orthogonality of filters utilized in prominent models, including ResNet, Swin, ConvNeXt, HorNet, and DiNAT. Their weights are obtained by training on ImageNet. The average absolute cosine similarity, serving as a measure of orthogonality, ranges from 0.046 to 0.072, indicating a high degree of orthogonality (<< 1.0, which indicates all filters are equal). This finding supports the introduced ESR.
> >
> > | Model        | ResNet-18 | ConvNeXt-T | Swin-T | HorNet-T | DiNAT-L |
> > | ------------ | --------- | ---------- | ------ | -------- | ------- |
> > | Abs. CosSim. | 0.067     | 0.072      | 0.06   | 0.069    | 0.046   |
> >
> > Besides, your comments are insightful and inspiring, we think how to adjust the equivalent sampling rate based on the orthogonality of the filter is a very interesting and important problem which worth further investigation.
> >
> >
> >
> > ### Q2: Typo in Section 3.2
> >
> > [Authors’ response]
> >
> > We appreciate your careful comments. We have corrected it and thoroughly reviewed the section to ensure the accuracy of the content and to prevent similar situations.

---

> ### Author Response · Authors · 2023-11-22
> **Response to Reviewer LE8t question 3-4**
>
> ### Q3: Incorrect figure reference.
>
> > “Aligning with the observations in Figure 1 that false responses and merging mistakes predominantly exist in areas with relatively low aliasing scores.” — Could you clarify how we are associating aliasing scores with a region of an image?
>
> [Authors’ response]
>
> Thank you for your careful comments. The reference to the figure was indeed incorrect. We have rectified this to *"Align with the observations in Figure **2**, where false responses and merging mistakes predominantly exist in areas with relatively low aliasing scores."*
>
> In Table 1 of the main paper, we provide evidence indicating that increasing the blur kernel leads to a reduction in the overall aliasing score. Consequently, there is a decrease in displacement error, while the occurrences of false responses and merging mistakes exhibit an upward trend. This aligns with the statistical findings presented in Figure 2 of the main paper, illustrating the distribution of three errors. In this figure, we observed that the aliasing score of false responses and merging mistakes is lower than that of displacement error, providing consistent support for our observations.
>
> We also have included comprehensive visualizations and analyses in the revised appendix (Figure 7 in the appendix). As the figure cannot be added in the comment box, we recommend referring to Section J of the revised appendix for a detailed visual representation.
>
>
>
> ### Q4: How does aliasing lead to a loss of object response
>
> [Authors’ response]
>
> Thank you for your inspiring comments. We provide visualizations in Figure 5 of the revised appendix. As the figure cannot be added in the comment box, for your convenience, we have also attached our description of the visualization below. Please refer to Section J and Figure 5 of the revised appendix for a detailed visual representation.
>
> In Figure 5 of the revised appendix, we have marked areas with high aliasing scores in the features with red boxes. Owing to the aliasing, the downsampled features exhibit a severe jagged phenomenon[2, 3], resulting in the degraded representation of object shapes and boundaries (see Figures 5(c) and (e) in the revised appendix). When downsampling increases to 4×, the response of some objects fades or is lost (see Figure 5(e) in the revised appendix). It is noteworthy that widely used state-of-the-art models, such as Swin Transformer and ConvNeXt, adopt a total downsampling stride of 32$\times$, potentially leading to an even more severe loss of response. After applying the proposed DAF to address the aliasing artifact, the jagged phenomenon is relieved (see Figures 5(d) and (f) in the revised appendix) and the response of objects still remains strong in the 4× downsampled features (see Figure 5(e) in the revised appendix). This indicates a strong correlation between the aliasing effect and a loss of object response.
>
> Recent work [2] also observes a similar phenomenon, where an increase in image noise raises the high frequency in features (leading to aliasing), resulting in deeper features losing the response of interested objects. Applying a low-pass filter (which can be regarded as an anti-aliasing filter) can alleviate this problem and recover the response of interested objects.
>
> [2] Qian S, Shao H, Zhu Y, et al. Blending anti-aliasing into vision transformer. Advances in Neural Information Processing Systems, 2021, 34: 5416-5429.
>
> [3] Zou, X., Xiao, F., Yu, Z., Li, Y., & Lee, Y. J. (2023). Delving deeper into anti-aliasing in convnets. *International Journal of Computer Vision*, *131*(1), 67-81.
>
> [4] Chen, L., Fu, Y., Wei, K., Zheng, D., & Heide, F. (2023). Instance Segmentation in the Dark. *International Journal of Computer Vision*, 1-21.

---

> > ### Comment · Reviewer_LE8t · 2023-12-03
> > **Response to the Authors**
> >
> > Thanks for the clarifications and additional results. The work indeed explores a critical problem. That being said, the proposed solution is more of a heuristic, limiting the technical novelty of the paper.  However, the proposed solution is effective empirically (marginal improvements). Considering all these, I want to keep my score.

---

### Official Review · Reviewer_qrZx · 2023-11-09

**Soundness:** 3 good
**Presentation:** 3 good
**Contribution:** 3 good
**Rating:** 8
**Confidence:** 4

**Summary:**

The paper proposes two novel de-aliasing filter (DAF) and frequency mixing (FreqMix) modules to alleviate aliasing degradation by accurately removing or adjusting frequencies higher than the Nyquist frequency. The paper observes three different wrong segmentation types potentially caused by aliasing (a) False response, (b) Merging mistake (c) Displacement. In addition, the paper designed a simple de-aliasing filter to precisely remove aliasing as measured by their aliasing score. Additionally, we propose a novel frequency-mixing module to dynamically select and utilize both low and high-frequency information. The work has really comprehensive ablation studies, and the induction logic is comprehensive. They have proved their proposed method surpasses strong baselines Mask2Former, PointRend.

**Strengths:**

1. The experiment of the paper is very comprehensive and solid. The authors provide analysis of (1) the relationship between blur kernel size with Boundary, Three type errors, and Aliasing score.  (2) Noise level in effect to aliasing. (3) their cut-off frequency in relationship with accuracy and aliasing. (4) In comparison with other anti-aliasing modules. (5) Show effectiveness with model scaling up. (6) In Comparison with SoTA segmentation model.

2. The paper introduces the concept of equivalent sampling rate for the Nyquist frequency calculation and proposes an aliasing score for quantitative measurement of aliasing levels.

**Weaknesses:**

Actually, I think from a research perspective, I believe the paper is valid and sound so I recommend accepting this paper. However, from a higher level of view, many traditional problems including anti-aliasing have been eroded by the current trend of Large Models. The effectiveness of more training data or advances in pre-trained model weights will shrink the marginal gain of those methods. Especially for the transformer-based architecture, the downsample operation is no longer max-pooling even adaptive-pooling. this potentially alleviates the aliasing problem itself.

**Questions:**

It would be really appreciated if the authors shared their thoughts on whether aliasing is still a problem if the backbone downsampling is fully adaptive in a transformer-based architecture.

---

> ### Author Response · Authors · 2023-11-22
> **Response to Reviewer qrZx**
>
> ### Q1: Discussion about aliasing in a transformer-based architecture.
>
> [Authors’ response]
>
> I sincerely appreciate your encouragement and valuable feedback. I would like to further discuss the question of whether aliasing remains a concern in a transformer-based architecture.
>
> We think aliasing is still a problem in existing transformer-based architecture. Taking the renowned Vision Transformer (ViT) as an example, ViT [1] tokenizes images by splitting them into non-overlapping patches, which are then fed into transformer blocks. The tokenization and self-attention operations performed on these discontinuous patch embeddings can be viewed as downsampling operations, introducing a potential side effect of aliasing. It is essential to note that this downsampling operation is virtually unavoidable due to the spatial redundancy nature of the image [2] and huge computational costs without downsampling (increasing by 256$\times$ without downsampling in ViT).
>
> Several existing studies have acknowledged this concern. A straightforward solution to alleviate aliasing is to increase the sampling rate. Similar trends are observed in vision transformers, where the use of overlapped tokens [3] and smaller patch sizes [4] contributes to improved performance. However, escalating sampling rates incur quadratic computational costs. Consequently, I hypothesize that integrating appropriate anti-aliasing filters into the 'attending' process could offer a viable solution. In fact, existing work has empirically explored blending anti-aliasing filters into the vision transformer, reporting observed improvements[5].
>
> In conclusion, aliasing persists as a potential concern in transformer-based architectures, and prior studies have endeavored to address it empirically by enhancing the sampling rate or integrating anti-aliasing filters into the attention mechanism. Our work represents a step forward in quantitatively assessing and addressing aliasing in contemporary models, supported by theoretical foundations. This issue still presents ample opportunities for further investigation and exploration.
>
> We appreciate your insightful questions that delve into the depth of our work and highlight its broad prospects.
>
>
>
> [1] Alexey Dosovitskiy, Lucas Beyer, Alexander Kolesnikov, Dirk Weissenborn, Xiaohua Zhai, Thomas Unterthiner, Mostafa Dehghani, Matthias Minderer, Georg Heigold, Sylvain Gelly, Jakob Uszkoreit, and Neil Houlsby. An image is worth 16x16 words: Transformers for image recognition at scale. In *ICLR*, 2021.
>
> [2]He, K., Chen, X., Xie, S., Li, Y., Dollár, P., & Girshick, R. Masked autoencoders are scalable vision learners. In CVPR, 2022
>
> [3] Li Yuan, Yunpeng Chen, Tao Wang, Weihao Yu, Yujun Shi, Zihang Jiang, Francis EH Tay, Jiashi Feng, and Shuicheng Yan. Tokens-to-token vit: Training vision transformers from scratch on imagenet. *in ICCV*, 2021.
>
> [4] Mathilde Caron, Hugo Touvron, Ishan Misra, Hervé Jégou, Julien Mairal, Piotr Bojanowski, and Armand Joulin. Emerging properties in self-supervised vision transformers.*in ICCV*, 2021.
>
> [5] Qian S, Shao H, Zhu Y, et al. Blending anti-aliasing into vision transformer. Advances in Neural Information Processing Systems, 2021, 34: 5416-5429.

---

### Author Response · Authors · 2023-11-22
**Summarization and common response**

## (1) Summarization

We are pleased that all reviewers acknowledge the innovative concepts from a research perspective and improved performance in experiments. The aliasing effect is a critical and attractive but under-explored topic ( `LE8t` and `uz5r`). This work provides a careful investigation of the aliasing phenomenon for semantic segmentation ( `uz5r`). The experiments are comprehensive and solid, surpassing a strong baseline ( `qrZx`). This research will inspire the related community ( `uz5r`).

(1) Despite the quantitative results, `LE8t` and `uz5r` point out that the experiment now lacks qualitative visualization to associate the aliasing effect with degradation. (2) `LE8t` requires a comparison of Pascal VOC and COCO,  (3) along with an analysis of time complexity.  (4) `qrZx` is interested in whether the aliasing effect also exists in the transformer-based model, while  (5)`LE8t` also asks for clarification on the applicable range of equivalent sampling rate.

In this revision, (1) we provide comprehensive visualized results (Figures 5, 6, 7, 8, and 9 in the appendix) to illustrate how the aliasing effect degrades features and how the proposed methods, DAF and FreqMix, effectively alleviate this degradation, thereby reducing three types of errors in the final predictions. (2) Additionally, we conduct supplementary experiments (Tables 4 and 5 in the appendix) on Pascal VOC and COCO, demonstrating the consistent improvement of our proposed methods across semantic segmentation (+1.8 mIoU on PASCAL VOC), object detection (+1.6 box AP on COCO), and instance segmentation tasks (+1.3 mask AP on COCO). (3) The revision includes a thorough analysis of time complexity (Table 4 in the appendix). (4) We confirm that the aliasing effect also exists in transformer-based models and discuss existing approaches to address it. (5) Furthermore, we discuss and justify the applicable range of the equivalent sampling rate by quantitatively analyzing the orthogonality of downsampling filters (Figure 4 and Table 3 in the appendix).

For your convenience, we have attached the revision details and changes to the PDF below. Please refer to the individual feedback for a detailed response.

## (2) Revision details

In this post, we enhance our manuscript by:

1. Providing visualizations to illustrate how aliasing degrades features and leads to three types of errors. Additionally, we offer comprehensive visualizations demonstrating how the proposed DAF and FreqMix address aliasing degradation in features and predictions, effectively reducing the occurrence of three types of errors ( `LE8t` and `uz5r`).
2. Presenting additional quantitative results on the PASCAL VOC and COCO datasets. The proposed method yields +1.8 mIoU on PASCAL VOC semantic segmentation, +1.6 box AP on COCO object detection, and +1.3 mask AP on COCO instance segmentation tasks ( `LE8t` ).
3. Providing time complexity analysis and showing that the inference speed is slightly slower (45ms to 49ms), excluding the impact of sub-optimal engineering implementations of frequency transformations, such as FFT/iFFT. Even with sub-optimal engineering implementations of FFT/iFFT, the training time only increases from 4.2 hours to 5.9 hours ( `LE8t` ).
4. Discussing the importance of the aliasing problem, even in large transformer-based architectures ( `qrZx` ).
5. Clarifying that the introduced equivalent sampling rate serves as a heuristic solution for selecting the cutoff frequency. It is based on the assumption of orthogonal downsampling filters in the models. We quantitatively analyze the orthogonality of downsampling filters and show high orthogonality in existing models, including ResNet, Swin, ConvNeXt, HorNet, and DiNAT, supporting the introduced equivalent sampling rate ( `LE8t`).

## (3) Changes to the PDF

**Main manuscript**

Only small fixes and wording improvements (marked as red in the manuscript):

- `[LE8t]` (Section 3.2) we revise the middle one to be the definition of MErr.
- `[LE8t]` (Section 3.2) we correct figure reference from*"aligning with the observations in Figure 1 …"*  to  *"aligning with the observations in Figure 2 …"*
- `[LE8t]` (Section 6) we add clarification about the equivalent sampling rate.

**Appendix**

The content added based on the reviews (marked as blue in the supplementary):

- `[qrZx]` Discussion about the aliasing problem in large transformer-based architectures (Section H).
- `[LE8t]` Time complexity analysis in Section I (Table 4).
- `[LE8t]` Orthogonality analysis of downsampling filters in Section H (Table 3).
- `[LE8t]` Additional results on PASCAL VOC and MS-COCO in Section L (Tables 4, 5).
- `[uz5r]` Visualized analysis for DAF and FreqMix in Sections J.1, J.2 (Figures 5, 6).
- `[uz5r]` Feature frequency analysis for DAF and FreqMix in Section K (Figures 8, 9).
- `[LE8t]` `[uz5r]` Visualized aliasing effect and how DAF and FreqMix improve feature and prediction in Section J.3 (Figure 7).

---

### Meta-Review · Area_Chair_CGce · 2023-12-06

**Metareview:**

This paper proposes anti-aliasing and frequency mixing to tackle the problem aliasing in segmentation. They provided a detailed case study and ablation study of the proposed approach. Overall, the writing is sufficient, the proposed approach is intuitive, achieved competitive results. Two reviewers recommend borderline rejects. However, I believe the most concerns have been addressed in the rebuttal and the paper tackles an interesting problem, albeit, less relevant these days due to large models.

**Justification For Why Not Higher Score:**

The topic may be of limited interest and the approach is more of a heuristic.

**Justification For Why Not Lower Score:**

I believe all the concerns were addressed. The paper is comprehensive and have sufficient contribution to the community.

---

### Decision · Program_Chairs · 2024-01-16

Accept (poster)